# Connected function of PRAF/RLD and GNOM in membrane trafficking controls intrinsic cell polarity in plants

Lu Wang[1,2,9], Dongmeng Li[1,2,9], Kezhen Yang[3,9], Xiaoyu Guo [1], Chao Bian [1,2,8], Takeshi Nishimura[4], Jie Le [3,5], Miyo Terao Morita [4], Dominique C. Bergmann[6,7] & Juan Dong [1,2✉]

Cell polarity is a fundamental feature underlying cell morphogenesis and organismal development. In the *Arabidopsis* stomatal lineage, the polarity protein BASL controls stomatal asymmetric cell division. However, the cellular machinery by which this intrinsic polarity site is established remains unknown. Here, we identify the PRAF/RLD proteins as BASL physical partners and mutating four *PRAF* members leads to defects in BASL polarization. Members of PRAF proteins are polarized in stomatal lineage cells in a BASL-dependent manner. Developmental defects of the *praf* mutants phenocopy those of the *gnom* mutants. GNOM is an activator of the conserved Arf GTPases and plays important roles in membrane trafficking. We further find PRAF physically interacts with GNOM in vitro and in vivo. Thus, we propose that the positive feedback of BASL and PRAF at the plasma membrane and the connected function of PRAF and GNOM in endosomal trafficking establish intrinsic cell polarity in the *Arabidopsis* stomatal lineage.

[1] Waksman Institute of Microbiology, Rutgers, The State University of New Jersey, Piscataway, NJ 08854, USA. [2] Department of Plant Biology, Rutgers, The State University of New Jersey, New Brunswick, NJ 08901, USA. [3] Key Laboratory of Plant Molecular Physiology, CAS Center for Excellence in Molecular Plant Sciences, Institute of Botany, Chinese Academy of Sciences, Beijing 100093, China. [4] Division of Plant Environmental Responses, National Institute for Basic Biology, Myodaiji, Okazaki 444-8556, Japan. [5] University of Chinese Academy of Sciences, Beijing 100049, China. [6] Department of Biology, Stanford University, Stanford, CA 94305, USA. [7] Howard Hughes Medical Institute, Stanford, CA 94305, USA. [8] Present address: Department of Plant Biology and Genome Center, University of California, Davis, Davis, CA 95616, USA. [9] These authors contributed equally: Lu Wang, Dongmeng Li, Kezhen Yang. ✉email: dong@waksman.rutgers.edu

Asymmetries in morphology and molecular distribution are fundamental features of the cells. Unevenly distributed, or "polarized", proteins are particularly critical for maintaining cellular structure and function in living organisms[1,2]. Asymmetric cell division (ACD) is a hallmark of stem cells that divide to self-renew while generating new cell types in the development of multicellular organisms. Stem cell ACD requires polarly localized protein complexes to regulate the asymmetries of division-plane placement and daughter-cell-fate determination[3,4].

In *Arabidopsis*, stomatal lineage cells are dispersed stem cells that undergo ACD to produce stomatal guard cells and pavement cells in the epidermis[5–7] (Fig. 1a and Supplementary Fig. 1a). During stomatal ACD, the plant-specific protein BASL defines an intrinsic polarity pole by asymmetrically distributing to the cell cortex[8] (Fig. 1a), where BASL assembles a polarity complex comprised of scaffold proteins (POLAR and BRX)[9,10] and signaling regulators (YODA kinase, BIN2 GSK3-like kinases, and BSL phosphatases)[11–13]. The loss-of-function of *BASL* results in

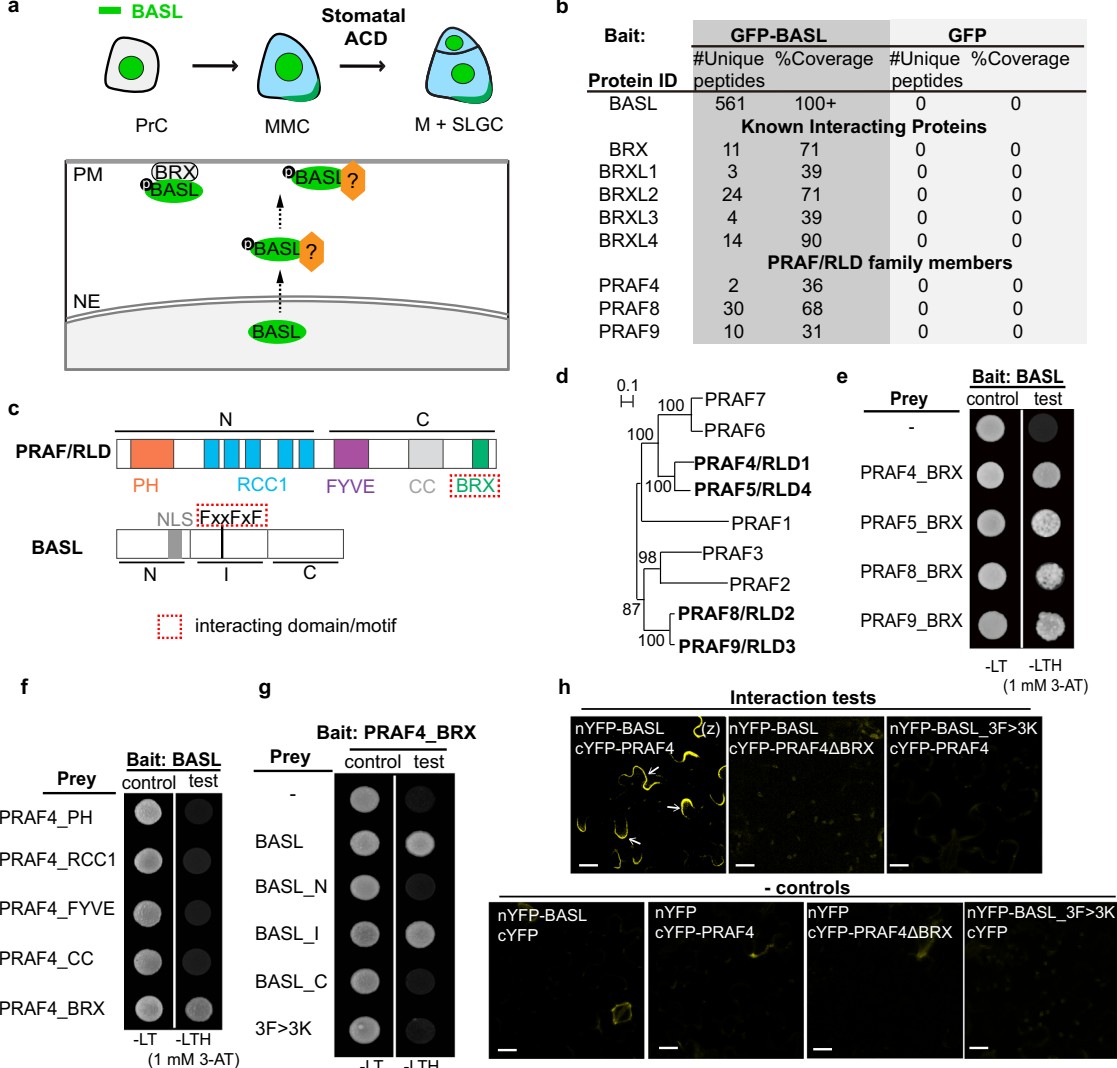

**Fig. 1 Identification of PRAF/RLD proteins as BASL physical partners. a** Localization of BASL (green) in *Arabidopsis* stomatal asymmetric cell division (ACD). PrC protodermal cell, MMC meristemoid mother cell, M meristemoid, SLGC stomatal lineage ground cell. Bottom schematic shows BASL polarization requires MAPK-mediate protein phosphorylation and the plasma membrane-associated BRX proteins. Unknown regulators (orange) are anticipated to mediate BASL polarization. PM, plasma membrane; NE, nuclear envelope. **b** Results of GFP-BASL co-immunoprecipitation (IP) coupled with mass spectrometry. The ubiquitous *35S* promoter was used to drive the expression of GFP-BASL in wild-type plants and proteins associated with GFP-BASL were isolated by GFP-trap agarose beads. GFP alone driven by the *BASL* promoter was used as control. For details, see Supplementary Data 1. **c** Subdomains of PRAF/RLD and BASL, respectively. "N", "I", and "C" stand for N-terminal, Internal and C-terminal domain of the protein, respectively. "NLS", nuclear localization signal. Dashed red boxes highlight identified interacting domain/motif. **d** Phylogenetic tree for the PRAF/RLD family, generated by Clustal W and based on the full-length proteins. Bold indicates proteins further characterized in this study. The AGI numbers can be found in "Methods". **e–g** Pairwise yeast two-hybrid assays. Bait, designated proteins fused with Gal4 DNA-binding domain (BD). Prey: "-" indicates Gal4 activation domain (AD) only, and the others are AD fused with designated protein or protein domain. 3F > 3K (BASL FxxFxF mutated to KxxKxK). "Test" indicates interaction assays on synthetic dropout media (-LeuTrpHis, -LTH); "Control" indicates yeast growth in rich media (-LeuTrp, -LT). When needed, specific concentrations of 3-Amino-1,2,4-triazole (3-AT) were supplied to suppress bait auto-activation. **h** Bimolecular fluorescence complementation (BiFC) assays in *N. benthamiana* leaf epidermal cells. Positive YFP signals indicate positive protein-protein interactions (BASL with PRAF4). Deletion of the BRX domain (PRAF4ΔBRX) or mutating the FxxFxF motif (BASL_3F > 3K) abolished YFP signals. White arrows show the interaction occurs in a polarized manner. Data represent the results of three independent experiments. Scale bars = 25 µm. (z), z-stacked confocal image.

compromised stomatal division and abnormal patterning of epidermal cells in *Arabidopsis*[8] (Supplementary Fig. 1a).

BASL is a peripheral membrane protein required for polarization of the other components in the polarity complex[9–12]. However, how BASL polarization is initiated and maintained in the stomatal lineage cells remains largely unknown. Previous studies showed that protein phosphorylation mediated by mitogen-activated protein kinases (MAPKs) and the GSK3-like kinases regulates BASL polarization and turnover[11,12]. Fluorescence Recovery After Photobleaching (FRAP) analyses that monitor protein intracellular dynamics revealed slow recovery of the BASL protein at the cell cortex, resembling that of an integral membrane protein, the PIN3 auxin efflux transporter in stomatal lineage cells[14]. It was thus hypothesized that the polarization of BASL protein is likely regulated by endomembrane trafficking and activities[15].

Directional auxin flows underlying developmental patterning and growth in plants is mediated by a network of auxin transporters, particularly the type I PIN effluxers, some of which are polarized to one side of the cell[16]. The PIN polarity maintenance heavily relies on the endomembrane trafficking system[16–19]. In these processes, an ADP-ribosylation factor guanine-nucleotide exchange factor (Arf GEF) GNOM switches on the activities of Arf small GTPases to promote endosomal recycling, thus the polar distribution of PIN1 at the plasma membrane[20–22].

Here, we identify a group of plant-specific PRAF/RLD proteins as physical partners of BASL and determine the essential role of PRAF/RLD for BASL polarization and stomatal ACD. We further find a direct functional connection between PRAF/RLD and the Arf GEF GNOM. We thus propose that the connected functions of PRAF/RLD with BASL and PRAF/RLD with GNOM underlie the intrinsic polarization of the peripheral membrane protein BASL in *Arabidopsis* stomatal lineage cells.

## Results

**PRAF/RLD proteins are physical partners of BASL.** To identify molecular components required for BASL polarization, we performed a genome-wide in vitro yeast two-hybrid screen and an in vivo co-immunoprecipitation (co-IP) assay coupled with mass spectrometry (MS). In the yeast two-hybrid, by using the full-length BASL protein as bait, 14 out of 17 positive interactions corresponded to a Brevis radix (BRX) domain that belongs to two plant-specific protein families, BRX[23,10] and PRAF[23,24]. In the in vivo co-IP experiments, plant materials used were the ubiquitous *35S* promoter-driven GFP-BASL expressed in wild-type *Arabidopsis* (detailed in Guo et al.[13] and in the "Methods" section). GFP-BASL proteins were extracted from 3-day-old seedlings and putative BASL-associating proteins were co-immunoprecipitated by GFP-trap agarose and analyzed by MS[13]. Consistent with the yeast two-hybrid results, candidate interacting proteins identified the BRX domain-containing proteins, including known partners, the BRX proteins[10], and putative partners, the PRAF proteins, i.e. PRAF4, PRAF8, and PRAF9 (Fig. 1b and Supplementary Data 1). The *Arabidopsis* genome contains 9 genes encoding the PRAF proteins, each of which is composed of multiple subdomains (Fig. 1c, d). Besides the BRX domain recognized for mediating protein-protein interactions[23], four additional subdomains include a Pleckstrin Homology (PH) domain, a cluster of Regulator of Chromosome Condensation 1 (RCC1) repeats, an FYVE (Fab1, YOTB, Vac1, and EEA1) zinc finger domain, and a Coiled-coil (CC) domain (Fig. 1c). The two phospholipid-binding domains (PH and FYVE) hint possible functional connection of PRAF with phospholipid signaling and membrane trafficking[25,26]. The CC domain could mediate protein-protein interaction and the RCC1 domain was found to

perform many functions, such as acting as a Guanine nucleotide exchange factor (GEF) to activate Ran GTPases[27,28]. Members of the PRAF family in *Arabidopsis* were recently named as RCC1-like domain (RLD) proteins that interact with the polarized LAZY1-like proteins and participate in the regulation of gravitropism signaling in root columella cells[29]. To characterize the function of PRAF/RLD proteins in BASL polarization and stomatal development, we focused on the four *PRAF* genes expressed in the vegetative tissues (based on the transcription profiling databases[30]), i.e., PRAF4/RLD1 (At1g76950), PRAF5/RLD4 (At5g42140), PRAF8/RLD2 (At5g12350) and PRAF9/RLD3 (At5g19420) (Fig. 1d).

In the pairwise yeast two-hybrid assays, we confirmed the physical interaction between BASL and the BRX domain of the four PRAFs (Fig. 1e). However, neither the full-length PRAFs nor the other individual subdomains of PRAF outside of the BRX domain showed detectable interaction with BASL in yeast (Fig. 1f and Supplementary Fig. 1b). We suspected that intramolecular domain folding of the full-length PRAF protein might interfere with BASL interaction, or that lipid-binding motifs would make PRAF translocation that makes the assays in the nucleus difficult in yeast. In vitro pull-down assays using *E. coli*-made recombinant proteins showed that BASL interacts with the carboxyl side of PRAF4, PRAF5, PRAF8, and PRAF9 (PRAF4/5/8/9_C, containing FYVE, CC, and BRX, Supplementary Fig. 1c). Furthermore, deletion of the RCC1 domain in the N-terminal half of PRAF4 (PRAF4ΔRCC1) allowed the detection of its interaction with BASL in yeast (Supplementary Fig. 1b), indicating the RCC1 domain might negatively influence the full-length PRAF to interact with BASL. On the other hand, when BASL subdomains (N, I, and C) were tested, the internal BASL_I domain was the only one exhibiting a positive interaction with PRAF4_BRX in yeast (Fig. 1g). Further assaying BASL_I allowed us to identify a small fragment containing a hydrophobic motif FxxFxF that once mutated to hydrophilic KxxKxK (BASL_3F > 3K), the interaction with PRAF4_BRX was disrupted in the yeast two-hybrid (Fig. 1g). Therefore, our data suggest that the physical contact between BASL and PRAF proteins occurs through the FxxFxF motif of BASL and the BRX domain of PRAF (Fig. 1c).

To test the BASL-PRAF interaction in plant cells, we performed the bimolecular fluorescence complementation (BiFC) assay in *Nicotiana benthamiana* epidermal cells. The BiFC assay is based on the reconstitution of an intact fluorescent protein (YFP) when two complementary non-fluorescent fragments are brought together by a pair of interacting proteins[31]. In the BiFC, we detected positive interactions between BASL and all four PRAF proteins, whilst the negative controls (nYFP or cYFP alone coupled with corresponding protein fusions) did not give observable signals (Fig. 1h). Interestingly, compared with the subcellular localization of individual proteins (YFP-BASL, cytoplasmic/PM and nuclear; YFP-PRAF, PM and cytoplasmic fine punctate. Supplementary Fig. 1d), the interaction of BASL with all four PRAFs commonly occur at the PM and in a highly polarized manner (Fig. 1h and Supplementary Fig. 1e). Such polarization events were also detected when BASL was co-expressed with other established components of the polarity complex[11–13]. PRAF4 and PRAF5, PRAF8 and PRAF9 are close homologous pairs (the amino acid sequences of PRAF4 and 5 are 70% identical, and PRAF8 and 9 are 85% identical). Thus, we often used PRAF4 and/or PRAF8 as representative members for various assays in this study. Deleting or mutating the interacting domain/motif of either side (PRAF4ΔBRX or BASL_3F > 3K) abolished the BASL-PRAF interactions in the BiFC (Fig. 1h), again indicating their interaction requires the BRX domain of PRAF and the FxxFxF motif of BASL, respectively. Taken together, we provided in vitro and in vivo data demonstrating that the PRAF proteins have the characters of being BASL physical partners.

**PRAFs are required for stomatal ACD**. To genetically characterize the biological function of PRAFs in vivo, we deployed the CRISPR/Cas9-mediated mutagenesis[32] to knock out *PRAF4, 5, 8,* and *9* simultaneously in *Arabidopsis*. In our experiments, each of the four *PRAF* genes was targeted by one guide RNA on one of the exons (Supplementary Fig. 2a). Among the second generation of transgenic plants (T2), we found dwarfed seedlings with dark green cotyledons segregating in some of the populations (Supplementary Fig. 2b). Genotyping results showed that these mutant plants are somatically chimeric (Supplementary Fig. 2c). For example, each of such two individual mutants (C6 and C8, Supplementary Fig. 2b) harbors homozygous premature termination mutations in *PRAF4* and *PRAF5*, also contains more than two different alleles of *praf8* and *praf9* mixed with the wild-type copies (detailed information in Supplementary Fig. 2c). These somatically mutated plants were named *praf4c;5c;8c;9c* that can rarely survive through the seedling stage. To create a null quadruple mutant, we first isolated a Cas9-free triple mutant *praf5c;8c;9c* that carries three homozygous premature termination mutations (sequences in Supplementary Fig. 2c). *praf5c;8c;9c* was then crossed with a T-DNA insertional null allele *praf4t* (SALK_067605, Supplementary Fig. 2a). The homozygous quadruple *praf4t;5c;8c;9c* mutants are phenotypically more severe than *praf4c;5c;8c;9c* (Supplementary Fig. 2b) but can be maintained as heterozygous *praf4t/+;5c;8c;9c*.

Close examination of the cotyledon epidermis showed that both quadruple mutants, *praf4c;5c;8c;9c* and *praf4t;5c;8c;9c*, produced extra numbers of stomatal lineage cells (Fig. 2a, b), the identity of which was verified by the expression of the stomatal lineage-specific receptor protein Too Many Mouths, TMM-GFP[33] (Supplementary Fig. 2d). Furthermore, the typical divisional asymmetry of the stomatal lineage cells (calculated as ratios of the smaller size A relative to the large size B, Fig. 2c) was disturbed by the *praf* quadruple mutations, to some extent mirroring what was observed in *basl* mutants (Fig. 2a, c)[8]. The disruptions of both physical asymmetry and cell-fate asymmetry were found in *basl* mutants. By using the expression of the late meristemoid marker MUTE as readout[34], which is usually only found in the small daughter in the wild type, was indeed identified in both daughter cells in *praf4t;5c;8c;9c* mutants, as in *basl-2* (Supplementary Fig. 2e). However, none of the single T-DNA insertional *praf* null mutants showed obvious stomatal defects (Supplementary Fig. 3a, b), suggesting the four *PRAF* genes are redundantly needed in the regulation of stomatal development. When the four *praf* mutations combined with the *basl* null, the quintuple mutants phenocopied the *praf* quadruple mutants in both plant growth and stomatal development (Fig. 2a–c), suggesting that BASL and its function in stomatal development might be one of the pathways that PRAFs regulate. Then, we introduced GFP-BASL in the *praf* quadruple mutants and found that the BASL polarization was clearly affected in both *praf4c;5c;8c;9c* and *praf4t;5c8c;9c* mutants (Fig. 2d, e and Supplementary Fig. 3c, d), suggesting the presence of the four PRAF proteins are required for BASL to polarize at the cell cortex. Furthermore, after mutating the PRAF-interacting FxxFxF motif of BASL, GFP-BASL_3F > 3K failed to polarize (Fig. 2f and Supplementary Fig. 3d) or to rescue *basl-2* mutant stomatal defects (brackets in Fig. 2f). Thus, we propose that the four PRAF proteins are required for stomatal ACD, at least partially through promoting BASL polarization in the stomatal lineage cells.

**PRAF proteins are localized to the plasma membrane, Golgi, TGN/EE, and endosomes**. To elucidate the in vivo subcellular localization of the PRAF proteins, we generated fluorescent-protein tagged PRAF proteins in *Arabidopsis*. In general, the *PRAF* genomic region containing the promoter was fused with YFP and introduced into the loss-of-function mutants for functional tests. We found that both native-promoter-driven PRAF4-YFP and PRAF8-YFP can rescue the growth phenotypes of *praf4t;8t-1;9t-1*[29] and *praf5c;8c;9c* mutants, respectively (Fig. 3a and Supplementary Fig. 4a). The differential orientation of the fluorescent tag was also tested for function by comparing PRAF4-YFP with GFP-PRAF4 in *praf4t;8t-1;9t-1* mutants. Results show that the two orientations display similar subcellular distribution and are equally efficient in complementation (Supplementary Fig. 4a, b). Thus, the N-terminal and C-terminal PRAF protein fusions were interchangeably used in this study.

In *Arabidopsis* leaves, the overall distribution patterns of YFP-tagged PRAF proteins were similar, all of which showed predominant association with the plasma membrane and formed endosome-like accumulations in the cytoplasm (Fig. 3b and Supplementary Fig. 4a–d). The plasma membrane association of PRAF4/8 was verified by the plasmolysis experiments, in which signals of PRAF4/8-YFP at the cell periphery retracted with the plasma membrane when detached from the cell wall (Supplementary Fig. 4e). The endosomal localization of PRAF became more evident when protein levels were expressed highly, such as native promoter-driven PRAF8 in mature guard cells (arrows in Fig. 3b) or overexpressed in the stomatal lineage cells (driven by the *TMM* promoter) (Fig. 3c and Supplementary Fig. 4d). Interestingly, when highly expressed in the stomatal lineage cells, PRAF8 became obviously polarized (arrows in Fig. 3d), whereas this polarization was lost in the absence of *BASL* (Fig. 3e and quantification in 3f). In addition, when co-expressed in *Arabidopsis*, GFP-BASL and mCherry-PRAF8 became highly overlapping. More specifically, the nuclear pool of BASL was diminished but enriched to the PRAF8-positive endosomes, whilst PRAF8 became highly polarized together with BASL at the plasma membrane (Fig. 3g). These data further supported the in vivo physical interaction of BASL and PRAF.

Because the PRAF proteins are localized to the plasma membrane and the endosome-like structures, to test the functional contribution of the plasma membrane pool, we expressed an N-terminally myristoylated PRAF8 (modification sequence in ref. [35]) in the loss-of-function mutants. Indeed, myr-PRAF8, compared to the wild-type PRAF8, was predominantly accumulated at the plasma membrane, and hardly formed endosomal puncta (Fig. 3h *vs.* 3b). Results showed that myr-PRAF8 partially complemented the dwarf phenotype of *praf5c;8c;9c* mutants (Fig. 3i), indicating that the plasma membrane pool of PRAF8 contributes significantly to its function.

To verify the endosomal-like localization of PRAF proteins, we used the styryl dye FM4-64 that intercalates into the plasma membrane, is then taken into the cells by endocytosis[36]. The *Arabidopsis* seedlings (3-day old) expressing YFP-tagged PRAF4/5/8/9 (driven by the native promoter) were incubated with 8 μM FM4-64 for 40-min. We found PRAF proteins colocalized with FM4-64 at the PM and partially overlapped with the FM4-64-positive puncta in the cytoplasm (Fig. 4a and Supplementary Fig. 5a), supporting the physical association of PRAF proteins with the endosomes. To test whether the PRAF-associated endosomes participate in the endocytic recycling pathway, we further treated the seedlings with Brefeldin A (BFA), an Arf GEF inhibitor that disturbs endomembrane trafficking, leading to the formation of the so-called "BFA-body" compartments that contain aggregated Golgi and trans-Golgi network (TGN)/early endosome (EE) membranes[22]. Recent study by Qi et al.[37] demonstrated that 30–90 μM BFA effectively induces the formation of similarly structured BFA bodies in *Arabidopsis* stomatal lineage cells. Our results showed that 60-min 70 μM BFA treatment, in the presence of a protein synthesis inhibitor, cycloheximide (CHX, 50 μM),

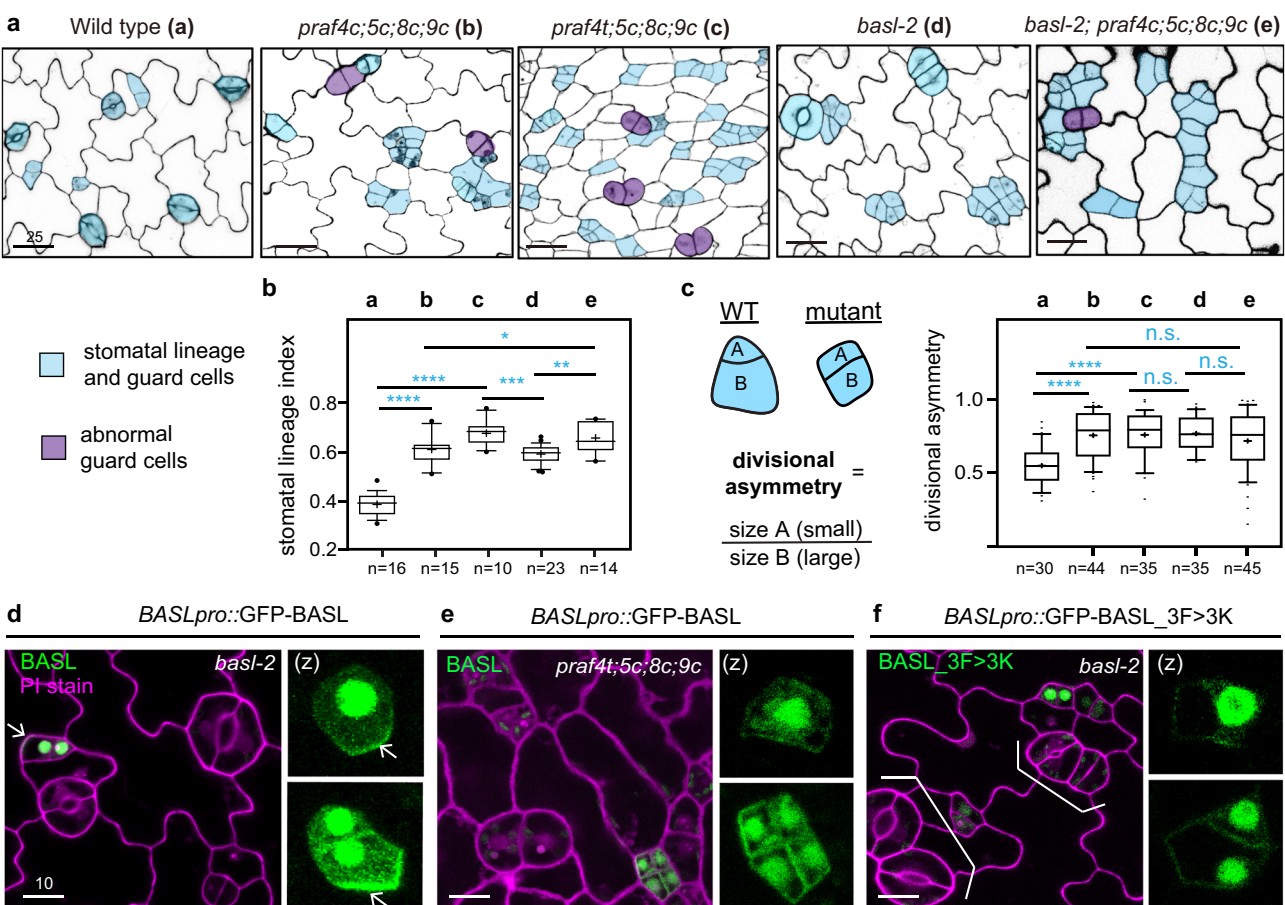

**Fig. 2 PRAF proteins are required for stomatal ACD and BASL polarization. a** Confocal images show abnormal stomatal division and differentiation in *praf* mutants. Five-day-old adaxial side cotyledon epidermis of the designated genotypes was examined. Cell outlines were visualized with Propidium Iodide (PI) staining and images were converted to black/white. Stomatal lineage cells were manually traced and highlighted by blue and abnormal guard cells were highlighted in purple. Data represent results of three independent experiments. Scale bars, 25 μm. **b–c** Quantification of stomata lineage index (**b**, ratio of # stomatal lineage cells/# total epidermal cells) and divisional asymmetry (**c**, ratio of cell sizes) for the genotypes shown in (**a**). Box plots show first and third quartile (box), median (line) and mean (cross). The parameters are applicable to all quantification data presented as box plots in this study. *n*, # cotyledons counted. Student's unpaired *t* tests were used for comparison. Two-sided *P* values are <0.0001 (for wild type vs. *praf4c;5c;8c;9c* and wild type vs. *praf4t;5c;8c;9c*), 0.0003 (for *basl-2* vs. *praf4t;5c;8c;9c*), 0.0027 (for *basl-2* vs. *basl-2; praf4c;5c;8c;9c*), and 0.0489 (for *praf4c;5c;8c;9c* vs. *basl-2; praf4c;5c;8c;9c*) in (**b**); and <0.0001 (for wild type vs. *praf4c;5c;8c;9c* and wild type *vs. praf4t;5c;8c;9c*), 0.7966 (for *basl-2* vs. *praf4t;5c;8c;9c*), 0.1664 (for *basl-2* vs. *basl-2;praf4c;5c;8c;9c*), and 0.3407 (for *praf4c;5c;8c;9c* vs. *basl-2; praf4c;5c;8c;9c*) in (**c**). n.s. not significant; *P < 0.05; **P < 0.005; ***P < 0.001; ****P < 0.0001. **d–f** Confocal images show disturbed GFP-BASL (green) polarization in *praf4t;5c;8c;9c* mutants (**e**) or upon the PRAF-interacting FxxFxF motif mutated (BASL_3F > 3K) (**f**). Data represent results of three independent experiments. Arrows in (**d**) indicate typical BASL polarization. White brackets in (**f**) show stomatal defects (clusters) typical for a *basl-2* mutant. PI staining was used to highlight cell outlines (magenta). Scale bars, 10 μm. (z), z-stacked confocal image. Quantification of BASL polarization in (**d–f**) is shown in Supplementary Fig. 3d.

triggered the aggregation of existing PRAF8 proteins around the BFA-body structures (Fig. 4a). Additional 2-h wash-out released the aggregation of PRAF8-YFP and recovered its original localization pattern (Fig. 4a). Similar responses were consistently observed for the other PRAF proteins (Supplementary Fig. 5a). Thus, the results suggested that cytoplasmic PRAF proteins associate with the endomembrane compartments that are sensitive to BFA in plant cells. On the other hand, we applied a selective phosphoinositide 3-kinases (PI3Ks) inhibitor, Wortmannin (WM), which disturbs late endosomal/vacuolar trafficking in plant cells. In the control experiment, the late endosomal marker YFP-RabF2a was dilated as anticipated (Supplementary Fig. 5b)[38]. In contrast, the WM treatment did not cause visible changes of the compartments decorated by Venus-PRAF8 in size or number (Supplementary Fig. 5b), suggesting the PRAF8-labeled compartments are not sensitive to defective vacuolar transport. Thus, our results revealed that the PRAF proteins partially associate with endomembrane

structures that are sensitive to BFA, such as the Golgi apparatus, TGN/EE, and/or other endosomal compartments[39].

To further define which endomembrane compartments the PRAF proteins associate with, we co-expressed fluorescent protein-tagged PRAF8 with the endomembrane WAVE and other maker lines[40] in *N. benthamiana* leaf epidermal cells. The markers we tested include the Golgi (G) markers, ST (a rat glycosyltransferase[41]) and Qb-SNARE MEMB12[40]; the TGN/EE markers, VHAa1[42], SYP61[43], and VAMP721[44], the post-Golgi endosomal markers (defined by[40]), RabC1, RabD1, RabD2a, and RabE1d; the endosomal/recycling endosomal (E/RE) markers[40], RabA1e and RabA5d; and the late endosomal/prevacuolar compartment (LE/PVC) markers, RabF2a and RabF2b[45,46]. The results show the most robust co-localization with PRAF8, based on the Pearson correlation coefficient (PCC) values[47], were RabC1- and RabE1d-decorated membrane compartments (Fig. 4b and Supplementary Fig. 6a, b).

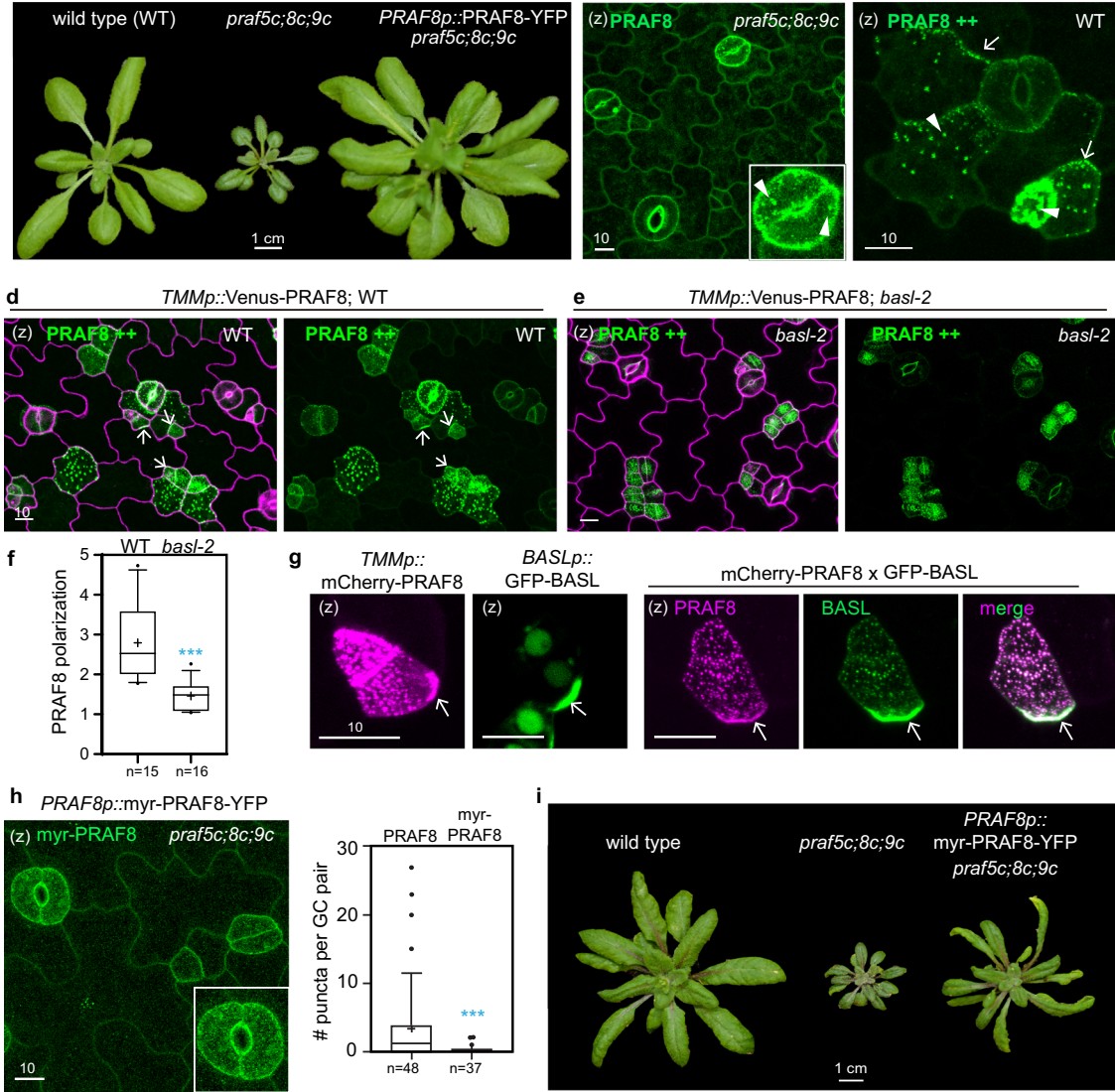

**Fig. 3 PRAF proteins are localized to the plasma membrane and endomembrane. a** YFP-tagged PRAF8 rescues growth defects of *praf5c;8c;9c*. Comparison of 4-week-old plants of the wild type, *praf5c;8c;9c*, and *PRAF8p*::PRAF8-YFP in *praf5c;8c;9c*. **b–c** Protein localization of PRAF8 (green) at the native level (**b**) or overexpressed in the stomatal lineage cells (PRAF8++, **c**) in 4-day-old adaxial side cotyledon epidermis. Inset in (**b**) shows an enlarged view of guard cells exhibiting endosomal localization of PRAF8 (arrowheads). **d–e** PRAF8 polarization (arrows) requires BASL. Overexpressed PRAF8 (*TMMp*::Venus-PRAF8, green) in wild type (**d**) and in *basl-2* (**e**), respectively. Data represent the results of three independent experiments. Magenta, PI-stained cell outlines. **f** Quantification of PRAF8 polarization in (**d, e**). Box plots show first and third quartile (box), median (line) and mean (cross). *n*, # stomatal lineage cells. Student's unpaired *t* tests were used. Two-sided *P* value is 0.0001. \*\*\**P* < 0.001. **g** Confocal images show individual protein expression and co-expression of *TMMp*::mCherry-PRAF8 (magenta) with *BASLp*::GFP-BASL (green). **h** Left, confocal image shows protein localization of *PRAF8p*::myr-PRAF8-YFP (green) in 4-day-old adaxial side cotyledon epidermis. Inset, enlarged view of guard cells. Note the reduced association of myr-PRAF8-YFP with intracellular particles compared to the wild-type PRAF8-YFP (inset in (**b**)). Right, quantification of # intracellular puncta for *PRAF8p*::PRAF8-YFP; *praf5c;8c;9c* and for *PRAF8p*::myr-PRAF8-YFP; *praf5c;8c;9c*, respectively. Box plots show first and third quartile (box), median (line) and mean (cross). n, # guard cell (GC) pairs counted. Student's unpaired *t* tests were used. Two-sided *P* value is 0.0005. \*\*\**P* < 0.001. **i** Expression of myr-PRAF8-YFP partially rescued the growth defects of *praf5c;8c;9c* mutants. Plants were about 4-week old. (z) stands for z-stacked confocal images. Scale bars in (**a**) and (**i**) are 1 cm and all others are 10 μm. Data represent results of three independent experiments.

The RabC1 structures have not been well characterized yet[48], whilst RabE1d was reported to localized to the plasma membrane and Golgi [49,50]. Partial co-localization of PRAF8 was also detected with the Golgi maker ST, the TGN/EE marker VAMP721, and LE/PVC RabF2b (Fig. 4b–d and Supplementary Fig. 6a, b). The association of PRAF8 with ST (Golgi) and VAMP721 (TGN/EE) appeared to be less stable or occur under certain conditions because only a portion of expressing cells showed positive co-localization (population a vs. b, Fig. 4c). In

summary, results of protein co-expression in *N. benthamiana* epidermal cells suggested that cytoplasmic PRAF8 partly associates with the Golgi, TGN/EE, and a subset of endosomal populations (Supplementary Fig. 6c).

**_praf_ mutants highly resemble _gnom_ mutants.** In the process of characterization of the *praf* mutant phenotypes, we noted that, strikingly, the developmental and growth defects of the *praf*

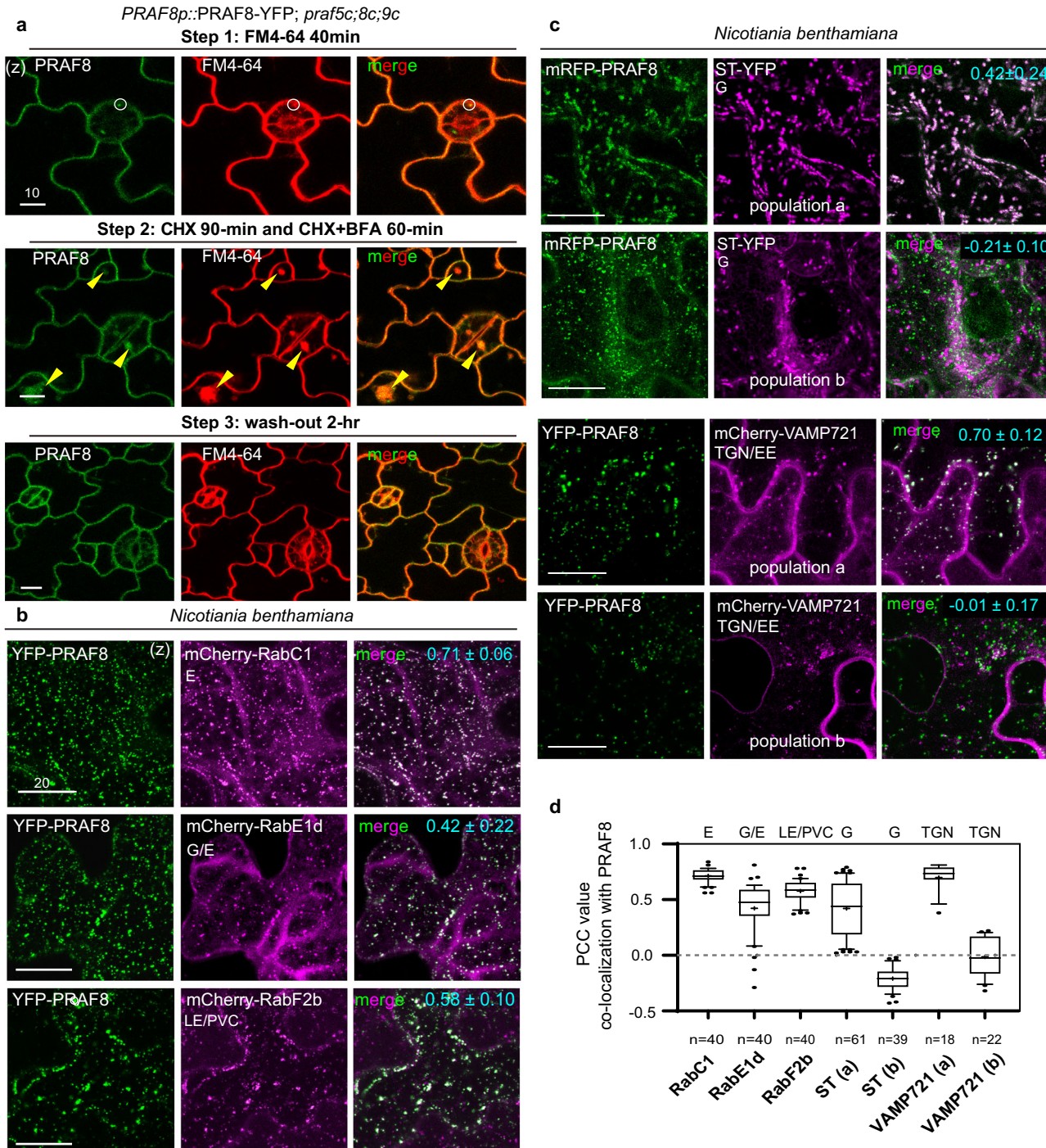

**Fig. 4 Intracellular PRAF8 partially associates with the Golgi, TGN/EE, and endosomes. a** PRAF8-YFP associates with the endosomes. Top panel: confocal images of PRAF8-YFP (green) in cotyledon epidermal cells stained with the endocytic tracer marker, 8 μM FM4-64 (red) for 40-min (step 1). White circles: overlapping endosomal signals. Middle, step 2, following FM4-64 staining, seedlings were treated with 50 μM cycloheximide (CHX, a protein synthesis inhibitor) for 90-min, followed by the addition of the Arf GEF inhibitor, 70 μM Brefeldin A (BFA), for 60-min. Step 3 (bottom), wash-out with water for 2 h. Yellow arrowheads mark the formation of "BFA-bodies" that overlap with PRAF8-YFP-positive aggregates. Data represent the results of three independent experiments. **b–c**. Co-localization of YFP/mRFP-PRAF8 (green) with mCherry/YFP-tagged endomembrane markers (magenta) in *N. benthamiana* leaf epidermal cells. Data represent results of three independent experiments. The co-localization rates (PCC, Pearson correlation coefficient values) are specified at the upper-right corner (cyan). G Golgi, E endosome, TGN/EE trans-Golgi network/early endosome, LE/PVC late endosome/prevacuolar compartment. **d** Quantification of protein co-localization based on PCC values in (**b**) and (**c**). For each pair of co-expression, *n* > 100 regions-of-interest (ROIs, each 228.3 μm²) were selected to obtain PCC values. ST showed 61/100 ROIs with PCC > 0 (population a), and VAMP721 showed 18/100 ROIs with PCC > 0 (population a). Box plots show the first and third quartile (box), median (line), and mean (cross). Data represent the results of three independent experiments. (z), z-stacked confocal image. Scale bars are 10 μm (**a**) and 20 μm (**b** and **c**).

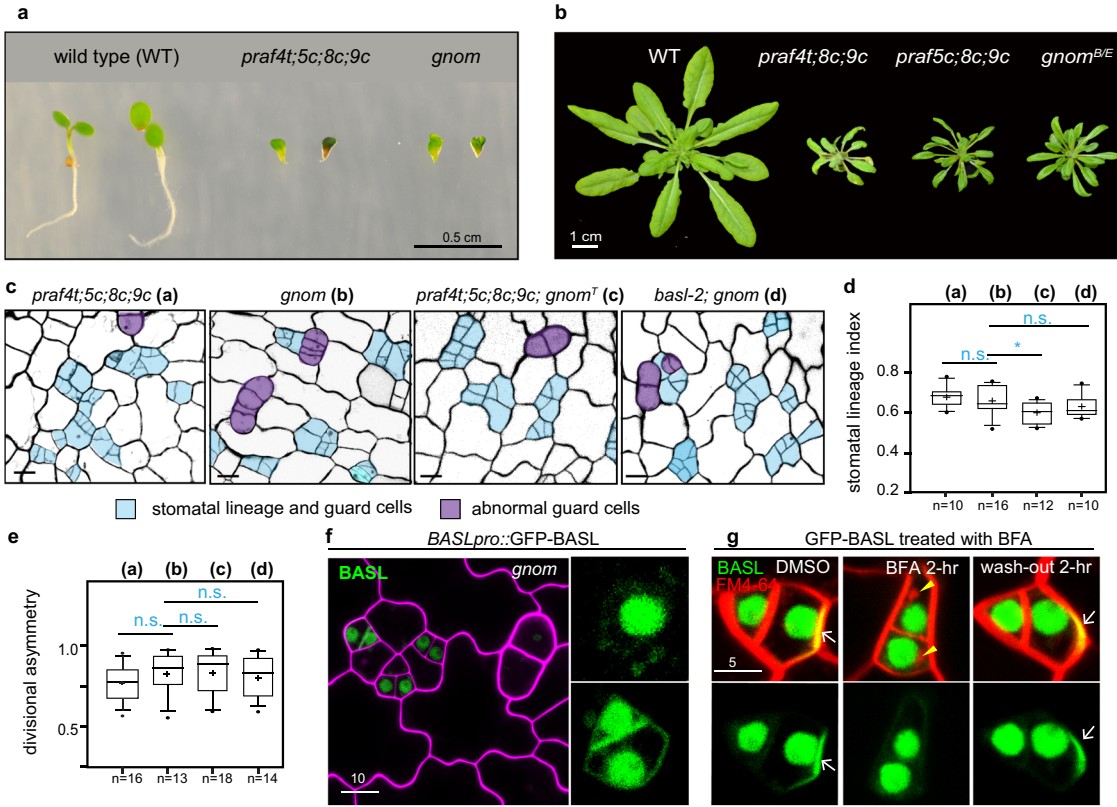

**Fig. 5 *gnom* mutants highly resemble *praf* mutants. a** 4-day-old seedlings of wild type, *praf4t;5c;8c;9c*, and loss-of-function *gnom* (segregated from *gnom^B/E*) mutants. **b** 4-week-old plants of wild type, *praf* triple mutants (*praf4t;8c;9c* and *praf5c;8c;9c*), and *gnom^B/E* (trans-heterozygously complementing *B4049* and *emb30-1* alleles). **c** Confocal images show stomatal phenotypes of 5-day-old adaxial cotyledon epidermis of the designated genotypes. Stomatal lineage and guard cells are manually traced and highlighted. Three independent experiments were performed. Scale bar = 10 μm. **d** Box plots show quantification of stomata lineage index for (**c**). Box plots show first and third quartile (box), median (line) and mean (cross). *n*, # cotyledons counted. Student's unpaired *t* tests were used for comparison. Two-sided *P* values are 0.4439 (for *gnom* vs. *praf4t;5c;8c;9c*), 0.0258 (for *gnom* vs. *gnom;praf4t;5c;8c;9c*), and 0.2922 (for *basl-2;gnom* vs. *gnom*). n.s. not significant; *P < 0.05. **e** Box plots show quantification of divisional asymmetry in the stomatal lineage for (**c**). Box plots show first and third quartile (box), median (line) and mean (cross). *n*, # daughter cell pairs. Student's unpaired *t* tests were used for comparison. Two-sided *P* values are 0.1707 (for *gnom* vs. *praf4t;5c;8c;9c*), 0.9274 (for *gnom* vs. *gnom;praf4t;5c;8c;9c*), and 0.6190 (for *basl-2;gnom* vs. *gnom*). n.s., not significant. **f** GFP-BASL (green) lost polarization in *gnom*. Magenta, cell outlines stained by PI. Data represent the results of three independent experiments. **g** Confocal images show GFP-BASL (green) merged with 8 μM FM4-64 staining (red) (top) or GFP-channel only (bottom). Three-day old seedlings were pre-incubated with FM4-64 for 40-min (left), then treated with 70 μM BFA for 2 h (middle), followed by 2 h wash-out (right). Yellow arrowheads indicate BFA-induced endomembrane aggregations. White arrows, protein polarization. Note no BASL polarization observed after BFA treatment (middle). Three independent experiments were performed. Scale bar, 5 μm.

mutants highly resemble those of the *gnom* mutants, as also noted by Furutani et al.[29]. GNOM is an activator of Arf small GTPases and functions in endomembrane trafficking to promote the recycling of internalized proteins, including the auxin-efflux carrier PIN1, to the basal plasma membrane[21,51]. The loss-of-function *gnom* mutants are defective in the directional transport of auxin, resulting in seedling-lethal phenotypes[21]. The loss-of-function *praf4t;5c;8c;9c* mutants replicated the strong *gnom* mutants (homozygous *B4049* or *emb30-1* alleles segregated from of a trans-heterozygotes *gnom^B4049/emb30-1* plant[52]) (Fig. 5a). Both mutants at 4-day old are dwarfed, produce fused cotyledons, and barely make any roots (Fig. 5a). Furthermore, the *praf* triple mutants, i.e. *praf4t;8c;9c* and *praf5c;8c;9c*, resembled the weak allele of *gnom^B/E* mutants (carrying trans-heterozygously complementing *B4049* and *emb30-1* alleles) and both *praf* triple and *gnom^B/E* mutants grow smaller and produce narrower rosette leaves (Fig. 5b). These highly resembling phenotypes hinted the possible functional connection between PRAF and GNOM in plant growth and development.

In stomatal development, *praf4t;5c;8c;9c* and *gnom* null mutants are also highly mirroring each other; both mutants produce elevated numbers of abnormal cell divisions (Fig. 5c and quantification in Fig. 5d, e) and both *praf* and *gnom* mutants produce strikingly similar, peanut-shaped guard cells[53] (Fig. 5c). Among the lower-order mutants, i.e. *praf4t;8t-1;9t-1* and *praf5c;8c;9c*, resembled the weak *gnom^B/E* mutants in producing mildly clustered stomatal lineage cells (Supplementary Fig. 7a). To assay the genetic interaction between *PRAF* and *GNOM*, we generated the quintuple null mutant *praf4t;5c;8c;9c;gnom^T*, in which a T-DNA insertional null allele of *gnom* was used[54]. We found that the stomatal phenotypes of a quintuple *praf4t;5c;8c;9c;gnom^T* mutant phenocopied that of the quadruple *praf4t;5c;8c;9c* or *gnom* mutants (Fig. 5c–e). Furthermore, *basl-2;gnom* was also indistinguishable from *basl-2;praf4c;5c;8c;9c* (Figs. 2a and 5c). Taken together, our phenotypic analyses revealed highly resembling phenotypes of the *praf* and *gnom* mutants in overall plant growth and stomatal development.

**GNOM is required for BASL polarization at the plasma membrane.** Because similar defects were observed in *praf* and *gnom* mutants, we suspected BASL polarity is also regulated by GNOM. Thus, we introduced the native promoter-driven GFP-

BASL into *gnom* mutants and found that consistent with what we observed for GFP-BASL in *praf* quadruple mutants (Fig. 2e and Supplementary Fig. 3c), *gnom* stomatal lineage cells largely failed to polarize BASL (Fig. 5f and Supplementary Fig. 3d), supporting that the activity of GNOM is required for BASL polarization. Furthermore, we treated 3-day-old seedlings expressing GFP-BASL with the Arf GEF inhibitor BFA and found BASL polarization was disturbed after 1 h and fully abolished by 2-h 70 µM BFA treatment (Fig. 5g and Supplementary Fig. 7b, quantification in Supplementary Fig. 3d). Washing out BFA to alleviate the inhibition of Arf GEFs enabled BASL to re-establish polarity within 1 h (Fig. 5g and Supplementary Fig. 7b). The results suggest that the Arf GEF-mediated trafficking is essential for BASL polarization.

**PRAF and GNOM proteins physically interact**. The highly resembling phenotypes of *praf* and *gnom* mutants inspired us to further investigate whether the PRAF and GNOM proteins directly interact. In the yeast two-hybrid assays, because the expression of full-length GNOM kills the yeast cells, we split GNOM into two halves (GNOM_N and GNOM_C, Fig. 6a), and no interactions were detected with the four full-length PRAF proteins (Supplementary Fig. 8a). We then tested whether GNOM_N or GNOM_C may interact with the subdomain of PRAFs. By using PRAF4 as a representing member, we tested PRAF4_PH-RCC1 (the fragment containing PH and RCC1) and PRAF4_FYVE-CC (containing FYVE and CC). Positive interactions were detected between GNOM_C with PRAF4_FYVE-CC but not with PRAF4_PH-RCC1, whereas no interactions were detected for GNOM_N (Fig. 6b). However, further narrowing down the PRAF4_FYVE-CC fragment did not allow us to detect interactions between GNOM_C with PRAF4_FYVE or PRAF4_CC (Fig. 6b). In the in vitro pull-down assays using recombinant proteins produced by *E. coli*, we further confirmed the physical association between GNOM_C and PRAF4_C that contains FYVE-CC and BRX (Supplementary Fig. 8b). Taken together, our in vitro data suggest that PRAF may physically interact with GNOM via the FYVE-CC subdomain.

Next, we tested the PRAF-GNOM interaction in plant cells by the BiFC assay in *N. benthamiana* epidermis. While the negative controls did not produce detectable signals, the complimented YFP signals for PRAF4/5/8-GNOM appeared as punctate compartments in the cytoplasm (Fig. 6c and Supplementary Fig. 8c, PRAF9 showed strong autoactivation in the BiFC, thus was excluded from the assay). To test the in vivo interaction of PRAF-GNOM, we performed co-IP experiment by co-expressing the ubiquitous *35S* promoter-driven GNOM-Myc together with the native promoter-driven YFP-tagged PRAF proteins in *Arabidopsis* plants. Because of the relatively low expression levels of PRAF4/5/8-YFP (Supplementary Fig. 8d), we relied on the plants expressing PRAF9-YFP. The co-IP results show that when PRAF9-YFP was pulled down by GFP-trap agarose beads, GNOM-Myc was identified to co-immunoprecipitate with PRAF9-YFP (Fig. 6d), supporting the physical association of GNOM with PRAF9 in vivo. To further visualize the PRAF-GNOM interaction in vivo, we examined the subcellular localization of GNOM-GFP alone (driven by the native promoter), mCherry-PRAF8 alone (driven by the *BASL* promoter), and the co-expression pattern when both proteins were present. Interestingly, when co-expressed, the two proteins became highly overlapping (PCC around 0.67) (Fig. 6e). Similar results were obtained when the two proteins co-expressed in *N. benthamiana* epidermal cells (Supplementary Fig. 8e, f). Lastly, we examined whether PRAFs and GNOM may impact each other's endogenous localization in *Arabidopsis*. Indeed, the native

promoter driven PRAF8-YFP or GNOM-GFP both showed abnormal aggregations in *gnom* or in *praf4t;5c;8c;9c* mutants, respectively (yellow arrowheads in Supplementary Fig. 9a, b), indicating PRAF and GNOM are mutually influential for their subcellular localizations. Thus, our biochemical, cell biological, and genetic data collectively supported that the PRAF and GNOM proteins may physically interact in vivo.

GNOM was determined to mainly localize to the Golgi apparatus in *Arabidopsis*[22]. Our results above suggested PRAF8 associates with the Golgi, TGN/EE, and endosomes (Fig. 4b–d and Supplementary Fig. 6a–c). We thus further investigate which subcellular compartments PRAFs may possibly interact with GNOM by expressing BiFC PRAF8-GNOM protein pairs with the mCherry WAVE markers[40] in *N. benthamiana* epidermal cells. Overall, the distribution of PRAF8-GNOM BiFC signals showed a mixed population of membrane structures with varying sizes (Fig. 6c and quantification of 3-D volume in Supplementary Fig. 10a). Aided by the WAVE markers, we detected the larger-sized vesicles ($10–110\ µm^3$) partially overlapped with RabE1d (Golgi and endosomes) and RabC1 (uncharacterized compartments) (white circles and insets in Fig. 6f), and no co-localizations were detected for other markers (Fig. 6f and Supplementary Fig. 10b). The results supported the possible interactions of PRAF8-GNOM may occur at the Golgi and RabC1-decorated structures in plant cells.

**praf mutants are defective in endomembrane trafficking**. The Arf GEF GNOM is well-characterized for its function in endosomal sorting and recycling of the auxin efflux carrier PIN1 at the PM[21,51]. We then tested whether *praf* mutants, as *gnom*, are defective in endomembrane organization and trafficking. We first examined the localization of the plasma-membrane protein PIN3 in stomatal lineage cells[53] (Fig. 7a and Supplementary Fig. 11a). Interestingly, in both *praf4t;5c;8c;9c* (*praf-quad*) and *gnom* mutants, the distribution of PIN3-GFP at the plasma membrane became less even (Fig. 7a, dashed arrows) and additionally accumulated to enlarged intracellular membrane compartments (Fig. 7a, yellow arrowheads), suggesting that transmembrane proteins at the plasma membrane undergoing endocytic recycling might be disturbed by the *praf* or *gnom* mutations.

The abundance and activity of plasma membrane proteins are regulated by endocytosis, endosomal sorting, endocytic recycling, and vacuolar degradation, etc.[55]. We first used FM4-64 internalization to measure endocytosis in the wild type, *praf-quad,* and *gnom* mutants. Results of FM4-64 internalization at sequential time points (5-, 10-, 20-, 30-, and 40-min treatment) suggested that the wild-type cells taking up FM4-64 was maximized at 15- to 20-min, whereas neither *praf-quad* nor *gnom* mutants showed obvious FM4-64 internalization until 30- to 40-min (white arrowheads in Fig. 7b and Supplementary Fig. 11b, quantification in Fig. 7c). Therefore, both PRAF and GNOM positively regulate endocytosis, consistent with the early study demonstrating the positive role of GNOM in endocytosis[56].

In the FM4-64 staining experiments, surprisingly, we identified a population of enlarged membrane aggregations in the cytoplasm of *praf-quad* and *gnom* mutants (yellow arrowheads in Fig. 7b and Supplementary Fig. 11b). These large compartments appeared very early when FM4-64 just became detectable in leaf epidermal cells (within 5-min) (Supplementary Fig. 11b) and the size and shape of these membrane aggregations resemble those of the "BFA-bodies" in wild-type plants (Fig. 7b and quantification in 7d). We suspect that the organization of the endomembrane system is disturbed in *praf-quad* and *gnom* mutants. To test this hypothesis, we examined a few YFP-tagged

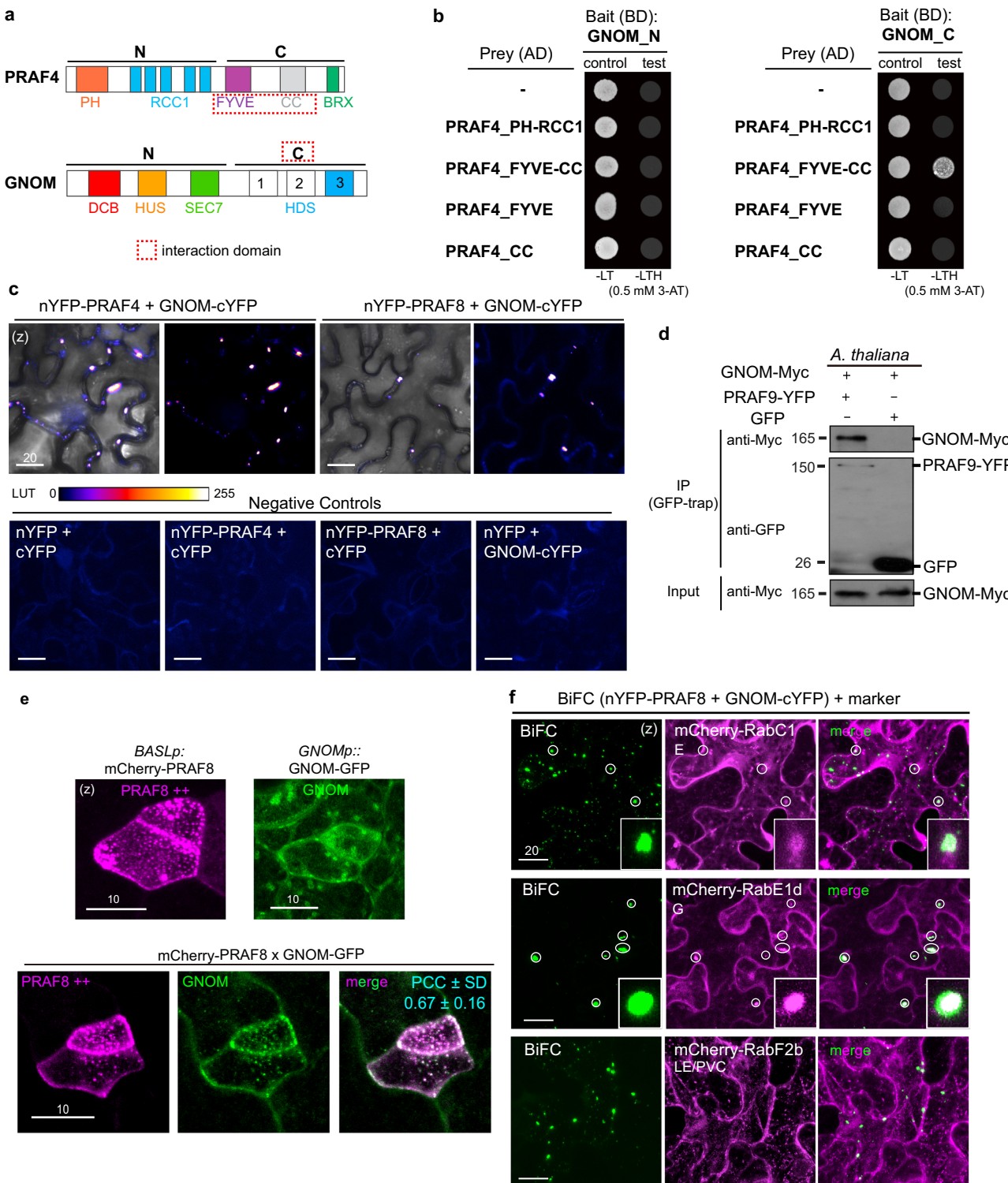

endomembrane makers, including RabE1d (Golgi/endosomal), endosomal RabD1, RabD2a, RabA1e, RabF2a, uncharacterized RabC1, and vacuolar VAMP711. Interestingly, most of these endosomal makers were more or less changed by *praf-quad* or *gnom* mutations and most of these changes were commonly found in both mutants (Fig. 7e and Supplementary Fig. 11c). Overall, the membrane compartments decorated by RabA1e, RabC1, RabD1, RabD2a, or RabE1d became smaller in both mutants, though the vacuolar marker VAMP711 remained unchanged in both mutants (Fig. 7e, f, and Supplementary

Fig. 11c). More specifically, in both *praf-quad* and *gnom* mutants, RabC1 appeared more diffused in the cytoplasm, RabD1 showed abundant accumulation at the cell cortical region, whilst RabE1d became more associated intracellular filamentous structures, likely the cytoskeletal elements (Fig. 7e and Supplementary 11c). Thus, our data suggested that PRAF and GNOM are required to maintain the morphology, organization, and function of many endomembrane compartments, in particular the Golgi, post-Golgi secretory and recycling pathways towards the plasma membrane in plant cells.

**Fig. 6 PRAF and GNOM proteins physically interact. a** Diagrams depict the domain structure of PRAF4 and GNOM, respectively. "N", the N-terminus; "C", the C-terminus. Dashed red box, identified PRAF-GNOM interacting domains. **b** Pairwise yeast two-hybrid assays show PRAF_FYVE-CC interacts with GNOM_C. Bait, GNOM_N (left) or GNOM_C (right) fused with BD. Prey, subdomains of PRAF4 fused with AD and "-" indicates AD only. "Test" means interaction assays on synthetic dropout media (-LTH). "Control" means yeast growth in rich media (-LT). Auto-activity of the bait protein fusions were suppressed by the addition of 3-AT. The result represents three biological repeats. **c** BiFC assays in *N. benthamiana* leaf epidermal cells show interactions of PRAF4 (left) or PRAF8 (right) with GNOM. nYFP N-terminal YFP, cYFP C-terminal YFP. Complemented YFP signals were converted to the ImageJ's Fire LUT mode. Data represent results of three independent experiments. **d** In vivo co-IP assays test the interaction between PRAF9 and GNOM. 5- to 7-day-old seedlings co-expressing *35S::GNOM-Myc* with *PRAF9p::PRAF9-YFP* or *BASLp::GFP* were used for the co-IP experiment. The numbers indicate protein sizes (kDa). The result represents three biological repeats. **e** Individual protein expression and co-localization of mCherry-PRAF8 (magenta) and GNOM-GFP (green) in stomatal lineage cells. Note the changes of GNOM alone vs. when co-expressed with PRAF8. Protein co-localization is calculated as PCC values (cyan). $n = 29$ stomatal lineage cells counted. **f** Co-localization of the BiFC PRAF8-GNOM interaction signals (green) with the WAVE endomembrane markers (magenta) in *N. benthamiana* leaf epidermal cells. Data represent the results of three independent experiments. Overlapping signals (white circles) were identified between PRAF8-GNOM BiFC with RabC1 (uncharacterized membrane compartments) and RabE1d (Golgi/endosomes) but not with the LE/PVC marker RabF2b. Insets show enlarged views of overlapping signals between BiFC and the WAVE marker. (z), z-stacked confocal images. Scale bars are as indicated (μm).

## Discussion

In this study, we identify members of the PRAF/RLD protein family as physical partners of the intrinsic polarity protein BASL in the stomatal lineage cells. Mutating the four *PRAF* genes (4, 5, 8, and 9) in *Arabidopsis* results in defects in cell polarity, stomatal lineage division, and general plant development. We further show that the PRAF proteins are localized to the plasma membrane, and partly associate with Golgi, TGN/EE, and post-Golgi endosomes in plant cells. The interaction between PRAF and BASL is required for BASL polarization at the plasma membrane in stomatal lineage cells. On the other hand, PRAFs become polarized in the presence of BASL in *Arabidopsis* stomatal lineage cells, suggesting a positive feedback relationship between BASL and PRAF for maintaining cell polarity. Furthermore, we show that PRAF protein physically interacts with the Arf GEF GNOM and *praf* and *gnom* mutants share common defects of disturbed protein distribution at the plasma membrane, abnormal endomembrane morphogenesis and trafficking in *Arabidopsis*. Thus, we propose that PRAF proteins function in the network of GNOM signaling to regulate endosomal trafficking to contribute to the cell polarity formation at the plasma membrane in plants (Fig. 7g).

The initial symmetry breaking of stomatal lineage cell appears to be intrinsic because the polarization axes of BASL crescents are largely random in a developing leaf in *Arabidopsis*[4,8] and BASL itself could spontaneously form a polarity site in cultured plant cells[57]. On the other hand, the BASL polarity is also regulated by external cues. Regeneration and reorientation of BASL polarity during reiterative stomatal ACDs are guided by both chemical and mechanical signaling-driven processes[58,59]. While previous work suggested protein phosphorylation-code serves as one of the mechanisms for BASL to polarize and function[11,12,14], our work here provides new mechanistic insights that endosomal activities mediated by the PRAF and GNOM proteins play a crucial role in establishing the BASL polarity domain in *Arabidopsis* stomatal lineage cells.

Polarization of membrane-associated proteins, such as the small GTPase Cdc42 in budding yeast and the PAR proteins in *C. elegans* zygote, requires positive feedback signaling loops[60]. In plants, GNOM as an activator of Arf GTPases is an endosomal regulator of vesicle budding and controls the basally (rootward) localized integral membrane PIN proteins[19,51]. In this study, we show that the PRAF proteins may function together with GNOM and regulate endosomal trafficking, thus cell polarization (Fig. 7g). Indeed, the polarized PIN proteins were found abnormally localized in same *praf/rld* quadruple mutants[29]. In a previous study that examined the polarization orientation of BASL protein in non-stomatal linage cells, Mansfield et al.[61] proposed a

common mechanism that defines a proximodistal field throughout the leaf epidermis may determine the polarization orientation of PIN1 and BASL. Here, our demonstration of GNOM and PRAFs required for BASL polarization was a surprise but indeed supports that, regardless of the cargo proteins being membrane integral (PIN1) or peripheral (BASL), GNOM and PRAF can promote both types of proteins to polarize at the plasma membrane.

In the cube-shaped root cells, opposing polarity domains are suggested by preferentially apical and basal localization of auxin transporters at the plasma membrane, such as apical AUX1, basal PIN1, and apical/basal PIN2 depending on cell types in *Arabidopsis*[19]. GNOM, possibly functioning with Arf1 and RabA members[62–64], plays a unique role in mediating endocytic recycling of the PIN proteins to the basal side[21]. The apical side polarization appears to be GNOM-independent and may require concerted activities of small RabA GTPases and the BIG clade Arf GEFs[65,66]. In the leaf epidermal cells, it remains unknown whether a complementary membrane domain opposing the BASL polarity site is actively maintained by certain landmark proteins in stomatal stem cells. Whilst intrinsic phosphorylation codes within the PIN proteins appear to direct their polarization orientation in a cell-type-specific manner[67–69], no evidence has been shown yet for BASL polarity orientation to flip to the other side caused by the differential phosphorylation status of the protein. The identified components of the polarity module, including the scaffolding proteins POLAR[9] and BRX families[10], the MAPKKK YODA[11], GSK3-like BIN2 kinases[12] and the BSL phosphatases[13], all overlap with the BASL crescent and require the presence of BASL for their polarization. It would be intriguing to test whether the compound endosidin 16 (ES16), which disturbs non-basal PM trafficking in the *Arabidopsis* root via interfering with the RabA GTPase-dependent pathway[66], has impacts on the BASL polarity module. Insights from such studies would further inform whether and how PRAF and GNOM contribute to the common regulatory theme underlying planar cell polarity in the leaf epidermis.

We demonstrated that the intracellular PRAF8 proteins are localized to the vesicular compartments that partially overlap with the Golgi, TGN/EE, and some endosomes in *N. benthamiana* leaf epidermal cells (Fig. 4 and Supplementary Fig. 6). Interestingly, the interactions between PRAF8 and GNOM were identified to mainly occur at the RabC1- and RabE1d-labeled endomembrane compartments (Fig. 6 and Supplementary Fig. 10), hinting the relevance or potential importance of the secretion/endosomal recycling in establishing cell polarity. RabC1 has not been well characterized in *Arabidopsis* yet, though its homolog Rab18 in mammals was found to associate with the

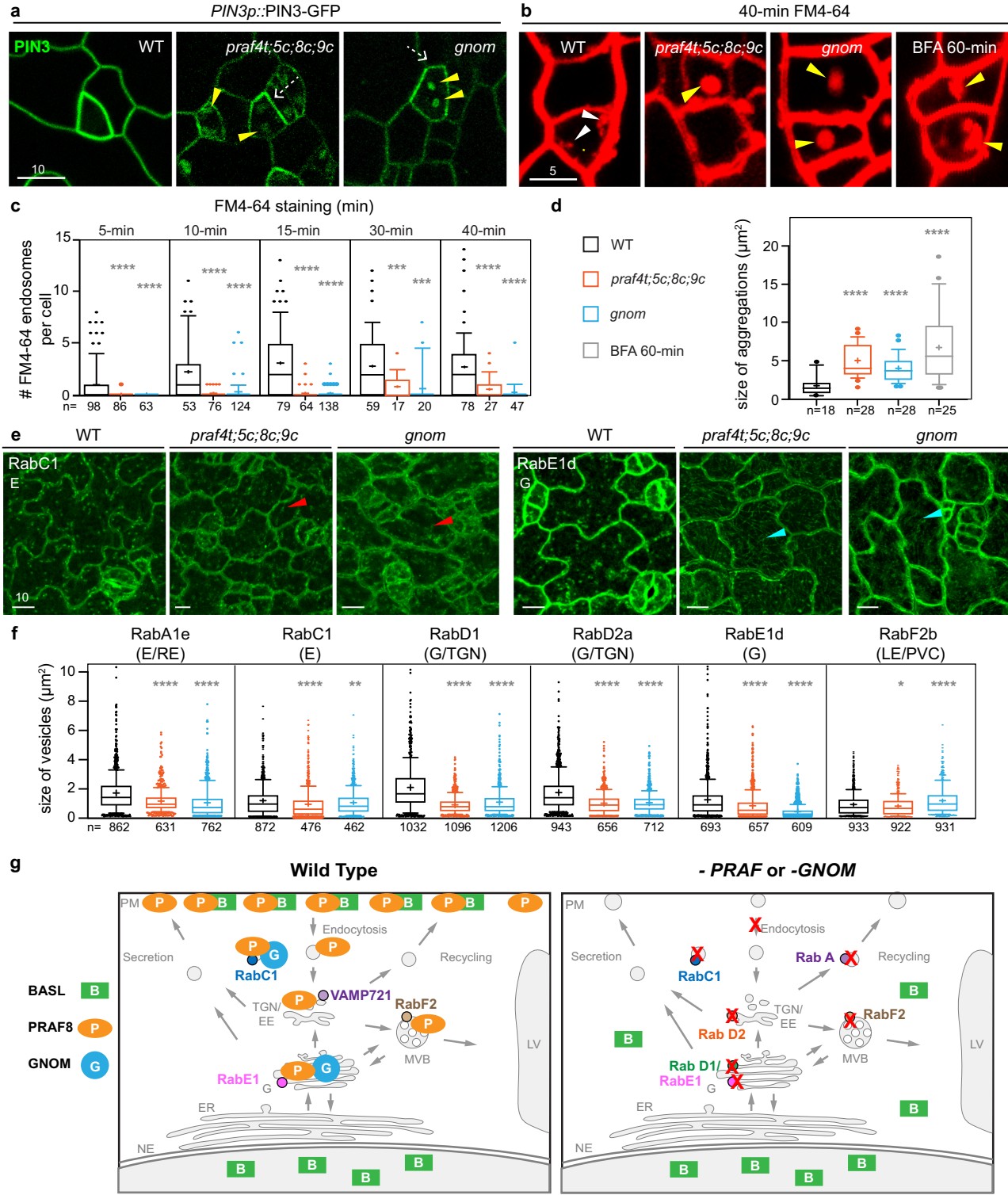

vesicles near the apical surface in polarized epithelial cells to promote targeted secretion[48,70]. The plant RabE GTPases, homologs of Sec4/Rab8 (yeast/mammals)[48], were found to localize to the Golgi and promote polarized exocytosis and secretion in both *Arabidopsis* and tobacco[49,50,71–73]. Thus, we propose that the connected function of PRAF-GNOM may regulate Golgi and post-Golgi endosomal activities, particularly the RabC- and RabE-mediated pathways, to promote directional exocytosis and secretion (Fig. 7g). This hypothesis is supported by the previous observation of cell wall defects in *gnom* mutants[74] and our

observation of abnormal morphogenesis and distribution of RabE1d and other organelle markers in both *gnom* and *praf* mutants (Fig. 7e and Supplementary Fig. 11c). Surprisingly, we did not detect a significant connection between RabA GTPases (RabA1e and RabA5d) and PRAF proteins (Supplementary Figs. 6 and 10b). The RabA GTPases are the homologs of Rab11 in animals[48,75] that function in secretion and endocytic recycling of PM proteins[45,65,76–79]. The *Arabidopsis* genome encodes an expanded RabA group comprising 26 members[80]. The possible functional connection between PRAF and RabA GTPases

**Fig. 7 PRAFs and GNOM are required for endomembrane trafficking. a** Localization of PIN3-GFP (green) in the designated genetic backgrounds. White dashed arrows indicate uneven distribution of PIN3 at the plasma membrane. Yellow arrowheads mark abnormal aggregations of PIN3 in the cytoplasm. Three independent experiments were performed. **b** FM4-64 (red) dye distribution (8 µM, 40 min) in stomatal lineage cells of WT, *praf4t;5c;8c;9c*, and *gnom* (segregated from *gnom^B/E*) to compare with WT treated with BFA (70 µM, 60 min). White arrowheads indicate internalized FM4-64 in WT. Yellow arrowheads indicate abnormal FM4-64 aggregations in *praf4t;5c;8c;9c*, *gnom*, or in WT treated with BFA. Data represent results of three independent experiments. **c** Quantification of endocytosis rates based on FM4-64 internalization. Box plots show numbers of FM4-64 positive endosomes *per* cell in WT, *praf4t;5c;8c;9c* and *gnom* after 5, 10, 15, 30, and 40 min of 8 µM FM4-64 treatment, respectively. Box plots show first and third quartile (box), median (line) and mean (cross). *n*, # cells measured. Student's unpaired *t* tests were used to compare with the wild type. Two-sided *P* values are 0.0001 (for WT vs. *praf4t;5c;8c;9c*) and 0.0004 (for WT vs. *gnom*) with 30-min FM4-64 treatment. All other two-sided *P* values are <0.0001. \*\*\**P* < 0.001; \*\*\*\**P* < 0.0001. **d** Quantification of sizes of FM4-64-positive compartments in designated backgrounds. Box plots show first and third quartile (box), median (line) and mean (cross). n, # FM4-64 positive compartments measured. Student's unpaired *t* tests were used in (**c**) and (**d**) to compare with the wild type. All two-sided *P* values are <0.0001. \*\*\*\**P* < 0.0001. **e** Localization of RabC1 (green, left) and RabE1d (green, right) in designated backgrounds. Red arrowheads mark more diffused RabC1 in the mutants. Cyan arrowheads mark filamentous distribution of RabE1d. Three independent experiments were performed. Scale bars in (**a**), (**b**), and (**e**) are as indicated (µm). **f** Quantification of vesicular sizes for RabA1e, RabC1, RabD1, RabD2a, RabE1d, and RabF2b in WT, *praf4t;5c;8c;9c*, and *gnom*, respectively. Box plots show first and third quartile (box), median (line) and mean (cross). *n*, # vesicles measured. Student's unpaired *t* tests were used. Two-sided *P* values are 0.0053 (for RabC1 vs. RabC1;*gnom*) and 0.0112 (for RabF2b vs. RabF2b;*praf4t;5c;8c;9c*). All other two-sided *P* values are <0.0001. \**P* < 0.05; \*\**P* < 0.005; \*\*\*\**P* < 0.0001. **g** Proposed working models for the subcellular localization of PRAF8 proteins (left) and trafficking pathways possibly interfered by the absence of the four *PRAF* genes (*PRAF4*, *PRAF5*, *PRAF8* and *PRAF9*) or *GNOM* (right). In wild-type plants, the polarization of BASL protein (green) in the stomatal lineage cell requires the physical partner, four PRAF proteins (orange), as well as the Golgi-localized Arf GEF GNOM (blue). The PRAF8 proteins are predominantly localized to the plasma membrane, where they may polarize together with BASL. The PRAF8 proteins may also partially associate with the Golgi, TGN/EE and a subset of endosomes/vesicles decorated by RabC1 and RabF2b. Furthermore, the PRAF proteins physically interact with GNOM, possibly leading to the association of GNOM to the RabC1- and RabE1d-decorated membrane structures. In the absence of the four *PRAF* genes or *GNOM*, multiple endomembrane markers are similarly defective in morphology and/or distribution (red crosses), suggesting that multiple routes, including endocytosis, secretion, and recycling etc., are commonly disturbed in *praf* and *gnom* mutants. We propose that the connected function of PRAF and GNOM plays important roles in endomembrane trafficking and is required for the establishment of BASL polarization in the stomatal lineage cells. PM plasma membrane, NE nuclear envelope, ER endomembrane reticulum, G Golgi, TGN/EE trans-Golgi network/early endosome, SV secretory vesicle, RE recycling endosome, MVB multivesicular body, LV lytic vacuole.

deserves more detailed investigation for their in vivo localization, genetic and biochemical interactions. Considering the fact that GNOM predominantly localizes to the Golgi apparatus[22], whilst PRAF8 localizes to both the Golgi, TGN/EE and post-Golgi endosomes, it is likely that PRAF proteins function as mediators of GNOM-driven Golgi-to-PM membrane trafficking that is required for the establishment of polarized membrane domains in *Arabidopsis*.

In responding to gravity signaling in columella cells of *Arabidopsis* lateral roots, the PRAF/RLD proteins become polarized to the plasma membrane by interacting with AtLAZY1/LAZY1-like (LZY) proteins. Polarized PRAF/RLD proteins may then direct PIN3 relocalization to modulate auxin flow for making changes in root growth angle[29]. In stomatal lineage cells, we showed that members of PRAF proteins interact with BASL and, when highly expressed in the stomatal lineage cells, become polarized in a BASL-dependent manner (Fig. 3d, e). Furthermore, myristoylated PRAF8 that is predominantly localized to the plasma membrane partially rescues mutant phenotypes, supporting the functional location of PRAF at least partly at the plasma membrane (Fig. 3h). This hypothesis is consistent with the previous finding that PRAF/RLDs interact with LZY to polarize at the PM to promote gravity signaling[29].

Then, what are the possible functions of PRAF at the plasma membrane in the stomatal lineage cells? Our hypothesis is that PRAFs might (1) through physical binding, stabilize BASL polarization at the plasma membrane, (2) promote regional cell expansion at the polarity site, and (3) regulate the PIN proteins and auxin signaling in stomatal lineage cells[53]. Considering the mild phenotype of the *pin* mutants in stomatal development (Supplementary Fig. 11d), it is likely the BASL polarity pathway was most significantly affected by the *praf* or *gnom* mutations. With regards to BASL polarization, the previously identified BASL partners, the BRX proteins, localize to the plasma membrane by palmitoylation, through which BASL can be stabilized at the plasma membrane[10]. The BRX domain in PRAFs can mediate

homotypic and heterotypic interactions between and within the BRX and PRAF family members[81,23], therefore PRAF may associate with the plasma membrane via its BRX domain to bind to BRX proteins and/or via its PH domain to bind to PtdIns(4,5)$P_2$. As GFP-BASL also requires PRAFs to polarize (Fig. 2d, e and Supplementary Fig. 3c, d), the interdependence of PRAF and BASL for polarization suggested a positive feedback loop between the two proteins for the establishment of cell polarity in stomatal lineage cells. Furthermore, when BASL is ectopically expressed, overaccumulation of BASL protein at the polarity site induces local cell expansion[8]. It is likely that PRAFs are recruited to the BASL polarity module, where PRAFs may crosstalk with phosphoinositide signaling, alter membrane property, and/or enrich regulators in membrane trafficking. The Rab GTPases in exocytosis and secretory pathways are promising candidates bridging polarized BASL and PRAF to enforced local deposition of cell wall materials in plant cells.

## Methods

**Plant materials and growth conditions**. The *Arabidopsis thaliana* ecotype Columbia (Col-0) was used as the wild-type unless otherwise noted. The sequence data of the proteins reported in this study can be found in TAIR (www.arabidopsis.org) with the following accession numbers: BASL (At5g60880), GNOM (AT1G13980), PRAF1 (At1g65920), PRAF2 (At3g47660), PRAF3 (At1g69710), PRAF4 (RLD1, At1g76950), PRAF5 (RLD4, At5g42140), PRAF6 (At3g23270), PRAF7 (At4g14370), PRAF8 (RLD2, At5g12350), and PRAF9 (RLD3, AT5g19420).

Arabidopsis mutants and marker lines used in this study were *basl-2*, GFP-BASL[8], TMM-GFP[33], MUTE markers[34,8], and *gnom^B/E* (a trans-heterozygously complementing line harboring *gnom^B4049* and *gnom^emb30-182*). The T-DNA insertional mutants were obtained from the Arabidopsis Biological Resource Center, ABRC, including *praf4t* (SALK_067605), *praf5t* (GABI_225B01), *praf8t* (SALK_061699), and *praf9t* (SALK_089136). The alleles of the triple mutant *praf4t;8t-1;9t-1* were described in ref. [29]. The GNOM-related reagents, including the endogenous promoter driven GNOM-GFP/RFP and *gnom^T* (SALK_103014) were reported in ref. [54]. The WAVE endomembrane markers (DNA and seeds) were described in ref. [40] and obtained from the ABRC.

To grow plants, in general, *Arabidopsis thaliana* seeds were surface sterilized and stratified for at least 1 day in the dark at 4 °C before transferred to the light. Seedlings were grown at 22 °C on half-strength Murashige & Skoog (MS) media in

1% agar plates supplied with 16-h light/8-h dark cycles for 6–10 days. Seedlings were then transferred to the soil for growth in a 22 °C growth chamber with 16-h light/8-h dark cycles. The wild-type *Nicotiana benthamiana* plants were grown at 25 °C and supplied with 14-h light/10-h dark cycles.

**Molecular cloning and transgenic plants**. In general, the LR Clonase II-based gateway cloning technology (Invitrogen) was used to generate constructs. All primer sequences are provided in Supplementary Table 1.

To generate C-terminal YFP-tagged PRAF protein fusion, the genomic coding regions of *PRAF* genes (4, 5, 8, and 9) (from ATG to immediately ahead of the stop codon) were amplified and cloned into pENTR/D-TOPO (Invitrogen), respectively. Next, the promoter regions were amplified and inserted into the *Not I* site of the pENTR/D-TOPO carrying genomic regions, respectively. To generate myr-gPRAF8-YFP, the myristoylation lipidation site was synthesized in the amplifying primers to attach to the genomic region of *PRAF8* for subcloning into pENTR/D. Then, the entry clone was recombined by the Clonase II enzyme mix (Invitrogen) into the destination vector pHGY for expression in plants[83].

To create N-terminal fluorescent-tagged Venus/mCherry-PRAF driven by the stomatal-lineage-specific *TMM* promoter, modified R4pGWB443 destination vectors (Tsuyoshi Nakagawa) containing the *TMM* promoter fused with Venus or mCherry tags were first created. The entry clones carrying genomic *PRAF* were recombined into the modified R4pGWB443 vectors containing the *TMM* promoter. To generate *PRAF4p::GFP-gPRAF4*, PCR product of *PRAF4* promoter was used to replace the *35S* promoter of pMDC43 vector via PmeI and KpnI sites. The entry clones carrying genomic *PRAF4* was recombined into this modified pMDC43 vector by a single LR reaction.

To express recombinant proteins in *E. coli*, the coding region of GNOM_C was cloned into pET28a vector to generate His-tagged GNOM_C. The coding region of PRAF4/5/8/9_C was cloned into pGEX-4T-1 vector to generate GST-tagged PRAF4/5/8/9_C. The construct expressing His-sumo-BASL was a gift from Tongda Xu Lab (Fujian Agriculture and Forestry University, China). To generate the plasmids used for protein co-localization and BiFC experiments in *N. benthamiana*, the full-length genomic *PRAF4*, genomic *PRAF4* with BRX domain deletion (PRAF4 ΔBRX), genomic *PRAF5*, genomic *PRAF8*, genomic *PRAF9*, coding sequence (CDS) of *GNOM* or *BASL* was first cloned in pENTR/D-TOPO, then recombined into pH35YG, pH35GY, pGWB641, pMDC43, pXNGW, pXCGW, pNXGW, or pCXGW[84]. The plasmids for expression of YFP- or mCherry-tagged WAVE marker lines were obtained from the ABRC. To generate site mutations in BASL_3F > 3K, the plasmid pENTR/D-TOPO carrying the CDS of *BASL* (-stop)[8] was used as template and specific point mutations were introduced by QuickChange II XL Site-Directed Mutagenesis Kit (Agilent) using specific primers listed in Supplementary Table 1.

The non-Wave makers used the BiFC co-expression assays in *N. bethamiana*, including ST-mRFP, mCherry-VAMP721, mCherry-SYP61, and VHAa1-mRFP, were created as described here. The coding sequences of ST[41] or VAMP721[44] were amplified and inserted into the *Not* I and *Asc* I sites of pENTR-D, followed by the LR reaction to make mRFP fusion driven by the *35S* promoter. The genomic sequences of VHAa1[42] or SYP61[43] were amplified and inserted into the *Not* I/*Asc* I sites of pENTR-D, followed by the LR reaction to make mCherry fusions driven the *UBQ10* promoter.

To express protein-of-interests in plants, destination vectors were introduced into *Agrobacterium tumefaciens* GV3101 and verified clones were infiltrated into *N. benthamiana* leaves[85] or transform *Arabidopsis* by the floral dipping method[86].

**Generation of *praf* mutants using CRISPR/Cas9**. Multiple sequence alignments using the full-length PRAF/RLD proteins were generated by Clustal W[87]. Similarity percentages between two proteins were obtained by BLASTP https://blast.ncbi.nlm.nih.gov/Blast.cgi?PAGE=Proteins. To generate CRISPR *praf4c;5c;8c;9c* mutant, the vectors (psgR-Cas9-At and p2xsgR-Cas9-At) and the protocol were deployed as described in ref. [32]. Specifically, each pair of the four PRAF guide RNAs (100 μM, Millipore Sigma) was phosphorylated and annealed following the manufacture's instruction (New England BioLabs, T4 polynucleotide kinase, M0201S). Oligo duplexes were then diluted 250 times and ligated in the *Bbs* I-digested psgR-Cas9-At plasmid[32]. To introduce the second guide RNA cassette, the pAtU6-PRAF5-sgRNA or pAtU6-PRAF9-sgRNA fragments were amplified with primers (pAtU6-F-KpnI and sgRNA-R-EcoRI) and ligated into pAtU6-PRAF4-sgRNA-Cas9 plasmid and pAtU6-PRAF8-sgRNA-Cas9 plasmid via *Kpn* I and *EcoR* I sites, respectively. Finally, the confirmed pAtU6-PRAF4-sgRNA-Cas9-pAtU6-PRAF5-sgRNA and pAtU6-PRAF8-sgRNA-Cas9-PRAF9-sgRNA cassettes were digested with *Hind* III and *EcoR* I and ligated into linearized pCAMBIA1300 and pCAMBIA2300, respectively. The plasmid pCAMBIA1300 containing PRAF4-sgRNA, PRAF5 CRISPR-sgRNA, and Cas9 was co-transformed with pCAMBIA2300 containing PRAF8-sgRNA, PRAF9-sgRNA, and Cas9 into the wild-type *Arabidopsis* simultaneously. A Cas9-free *praf5c;8c;9c* triple mutant was screened out by PCR-based genotyping from a somatically mutated *praf4c;5c;8c;9c* population. To introduce a marker line or another mutation into *praf4t;5c;8c;9c* mutants (seedling lethal), in general the new marker or mutant was first crossed with *praf5c;8c;9c*, followed by the progeny crossed with *praf4t*. Desired genetic materials were then screened out from the F2 populations.

**RNA extraction and RT-PCR**. Total RNAs were extracted from the whole *Arabidopsis* seedlings (4- to 5-day-old) using a RNeasy Plant Mini Kit (Qiagen). The first-strand cDNAs were synthesized by the qScript cDNA SuperMix (Quantabio) with 500 ng of total RNAs as template and oligo dT as primer. For reverse transcription (RT)-PCR, the ribosomal *S18* (*RPS18*) gene was used as an internal standard for normalization of gene expression levels. *PRAF* genes and *S18* were amplified for 28 and 25 cycles, respectively, with the primers listed in Supplementary Table 1.

**Chemical treatment**. Brefeldin A (BFA, Millipore Sigma) treatments were performed on 3- to 4-day-old *Arabidopsis* seedlings, grown as described above. Seedlings were submerged in the half-strength MS liquid medium containing 70 μM BFA (dissolved in dimethylsulfoxide, DMSO) or corresponding amount of DMSO (mock) in the light at room temperature. Treated seedlings after certain period of time, as described in the main text, were mounted for confocal imaging. FM4-64 (Synaptored C2, Biotium) labeling of BFA bodies involved a treatment of the seedlings in 8 μM FM4-64 for 40 min, followed by a treatment with 70 μM BFA or DMSO (mock). The Cycloheximide (CHX, Millipore Sigma) treatment was performed by submerging 4-day-old seedlings first in 8 μM FM4-64 for 40 min, followed by 50 μM CHX (0.1% DMSO) treatment for 90-min, and then followed by a treatment with 70 μM BFA and 50 μM CHX or DMSO. For BFA wash-out experiments, seedlings were then washed in ddH$_2$O and Confocal imaged post washing. For FM4-64 internalization experiments, 3-day-old seedlings were incubated for 5–40 min supplemented with 8 μM FM4-64 before mounting for imaging. For wortmannin treatment, 3- to 4-day-old seedlings were treated with water containing 33 μM wortmannin (Cayman Chemical) or corresponding amount of DMSO (mock) for 2 or 3 h. For the plasmolysis experiments, 4-day-old seedlings were first incubated with PI for 10-min and then 20% sucrose for 30-min prior to mounting for imaging.

**Yeast two-hybrid assay**. A genome-wide yeast two-hybrid screen for BASL physical interactors was performed as described in ref. [10]. The full-length BASL coding sequence cloned into pGBKT7 was used as bait to isolate interacting peptides expressed by a cDNA library derived from 3-day-old etiolated *Arabidopsis* seedlings (ABRC stock CD4-22).

For pairwise yeast two-hybrid tests, the coding sequences of related genes in pENTR/D were recombined into the bait vector pGBKT7 or the prey vector pGADT7 (Clontech). The EZ-Transformation Kit (MP Bio-medicals) was used for yeast transformation following the manufacturer's instructions. Bait and prey clones were co-transformed into the yeast strain AH109 and positive transformants were selected with SD/-Leu/-Trp medium after 2 days of yeast growth at 30 °C. The interactions were tested on SD/-Leu/-Trp/-His medium and observed after 3 days of yeast growth at 30 °C. To inhibit self-activation of certain protein fusions with the DNA-binding domain (BD), specific concentrations of 3-Amino-1,2,4-triazole (3-AT) was used as indicated in the figures.

**Transient protein expression, co-localization, and BiFC in *Nicotiana benthamiana***. The vectors and protocol of using *N. benthamiana* epidermal cells for transient protein expression, co-localization and Bimolecular fluorescence complementation (BiFC) assays were described previously in[85,88]. Specifically, prior to leaf infiltration, the *Agrobacterium tumefaciens* GV3101 harboring the expression vector was cultured overnight in 10 ml of Luria-Bertani (LB) medium containing appropriate antibiotics. Bacterial cells were then harvested at $4000 \times g$ for 10 min and resuspended in 10 ml of 10 mM MgCl$_2$, followed by another step of 10 mM MgCl$_2$ washing. Cells then remained in the medium for 3 h at room temperature prior to infiltration. Equal volumes of cell culture expressing the protein-of-interest and the strain expression the p19 protein (to suppress gene silencing)[89] were mixed to reach an optical density 600 (OD600) of 0.5 and infiltrated into the 4-week-old *N. benthamiana* leaves. Three to five days after infiltration, leaf disks were excised and mounted onto slides for confocal imaging.

**Recombinant protein production and pull-down assay**. For recombinant protein expression and purification, the recombinant BASL protein tagged by His-sumo and His-tagged GNOM_C were purified using Ni-NTA agarose (QIAGEN) according to the manufacturer's protocol. To perform the pull-down assay, GST or GST-PRAF_C proteins were immobilized on Pierce™ Glutathione Superflow Agarose (Thermo Scientific™), which were then incubated with equal amount of purified His-sumo-BASL or His-GNOM_C, respectively, on a rotating wheel at 4 °C for 4 h with gentle shaking. Then, the beads were collected and washed with the washing buffer (50 mM Tris-HCl pH 7.4, 150 mM NaCl, 0.5 mM EDTA, 0.5% TritonX-100) for 3–5 times to eliminate nonspecific bindings. The bound proteins on the Glutathione Agarose were then boiled and separated by 10% sodium dodecyl sulfate-polyacrylamide gel electrophoresis (SDS–PAGE) and transferred onto a polyvinylidene fluoride membrane (Bio-Rad). Proteins were analyzed by Immunoblot with Anti-His (His-Tag Antibody #2365, Cell Signaling Technology, 1:1000) or anti-GST (GST (91G1) Rabbit mAb #2625, Cell Signaling Technology, 1:1000) primary antibodies, respectively, and anti-mouse IgG, HRP-linked (#7076, Cell Signaling Technology, 1:1000), or anti-rabbit IgG, HRP-linked (#7074, Cell

Signaling Technology, 1:1000) secondary antibodies, respectively. The pull-down assays were repeated at least 3 times to acquire the representative image.

**Co-IP and MS.** To identify BASL-interacting proteins through co-IP MS, 5 g of seedlings (expressing 35S::GFP-BASL in Col-0, or BASL::GFP in Col-0 at 3-dpg) were ground to a fine powder in liquid nitrogen. Proteins were extracted with the extraction buffer (100 mM Tris-HCl at pH7.5, 150 mM NaCl, 5 mM EDTA, 5 mM EGTA, 10 mM DTT, 10 mM Na3VO4, 20 mM NaF, 50 mM β-glycerophosphate, 10% glycerol, 1 mM PMSF, protease inhibitor cocktail (Millipore Sigma, P 9599), 1% (v/v) NP-40). The homogenates were sonicated for 10 s, then diluted to NP-40 <0.2% (v/v) in the protein extracts. After centrifuge at $10,000 \times g$ for 30 min at 4 °C, the supernatant was transferred to a new tube to mix with 100 μl GFP-Trap Agarose (Chromotek) and incubate for 3 h on a rotating wheel at 4 °C. The beads were then collected by low-speed centrifugation, followed by 4 times of wash with the extraction buffer with 0.2% (v/v) NP-40. Finally, 5× SDS sample buffer was added to the beads and boiled at 95 °C for 5 min. Protein samples were separated by 10% SDS–PAGE by a short distance, followed by gel reduction, alkylation, and digestion with trypsin (sequencing grade, Thermo Scientific Cat # 90058). Peptides were extracted twice with 5% formic acid, 60% acetonitrile, and dried under a vacuum.

LC-MS/MS analysis was performed at the Biological Mass Spectrometry facility of Rutgers University. Samples were analyzed by LC-MS using Nano LC-MS/MS (Dionex Ultimate 3000 RLSC nano System) interfaced with QExactive HF (Thermo Fisher). Peptides were loaded on to a fused silica trap column Acclaim PepMap 100, 75 μm × 2 cm (Thermo Fisher). After washing for 5 min at 5 μl/min with 0.1% TFA, the trap column was brought in-line with an analytical column (Nanoease MZ peptide BEH C18, 130 A, 1.7 μm, 75 μm × 250 mm, Waters) for LC-MS/MS. Peptides were fractionated at 300 nL/min using a segmented linear gradient 4–15% B in 30 min (where A: 0.2% formic acid, and B: 0.16% formic acid, 80% acetonitrile), 15–25% B in 40 min, 25–50% B in 44 min, and 50–90% B in 11 min.

MS data were acquired using a data-dependent acquisition procedure with a cyclic series of a full scan acquired in Orbitrap with resolution of 120,000 followed by MS/MS (HCD relative collision energy 27%) of the 20 most intense ions and a dynamic exclusion duration of 20 s. The peak list of the LC-MS/MS were generated by Thermo Proteomoe Discoverer (v. 2.1) into MASCOT Generic Format (MGF) and searched against Arabidopsis (TAIR v. 10), plus a database composed of common lab contaminants using an in house version of X!Tandem (GPM Furry, Craig and Beavis, 2004). Search parameters are as follows: fragment mass error: 20 ppm, parent mass error: ±7 ppm; fixed modification: carbamidomethylation on cysteine; flexible modifications: Oxidation on Methionine; protease specificity: trypsin (C-terminal of R/K unless followed by P), with 1 miss-cut at preliminary search and 5 miss-cut during refinement. Only spectra with loge<−2 were included in the final report.

To test in vivo physical association between GNOM and PRAF, total cell proteins were extracted from 5- to 7-day-old seedlings co-expressing 35S::GNOM-Myc and PRAF9pro::PRAF9-YFP. The negative control was performed with proteins extracted from plants co-expressing 35S::GNOM-Myc and BASLpro::GFP. Plant tissues were first grounded up in liquid nitrogen to a fine powder and mixed with the protein extraction buffer (100 mM Tris-HCl pH 7.5, 5 mM EDTA, 5 mM EGTA, 1 mM Na3VO4, 10 mM NaF, 50 mM b-glycerophosphate, 10 mM DTT, 1 mM phenylmethylsulfonyl fluoride, 5% (v/v) glycerol, 0.5% (v/v) Triton X-100, and 1% (v/v) protease inhibitor cocktail (Millipore Sigma, P 9599)). The mixture was then incubated with the GFP-Trap Agarose beads (Chromotek, pre-washed by 1 mL extraction buffer) for 4 h at 4 °C. Then, the beads were collected by spinning and washed with the extraction buffer for 3–5 times to eliminate nonspecific bindings. The bound proteins on the GFP-trap were boiled and assayed by 10% SDS–PAGE followed by immunoblotting with corresponding primary antibodies, i.e. anti-GFP (Anti-GFP #11814460001, Roche, 1:1000) or anti-Myc (Myc-Tag (9B11) Mouse mAb #2276, Cell Signaling Technology, 1:1000), and secondary antibodies (Anti-mouse IgG, HRP-linked Antibody #7076, Cell Signaling Technology, 1:1000).

**Confocal imaging, data processing, and quantification analysis.** Confocal images were captured with the Leica TCS SP5 II Confocal microscope (40× or 63×). Unless otherwise noted, the adaxial epidermis from cotyledons of 3- or 4-day-old seedlings were used for stomatal phenotyping. Cells outlines were visualized by the propidium iodide (PI, Invitrogen) staining. The excitation/emission spectra for various fluorescent proteins are as follows: GFP, 488 nm/501–528 nm; YFP/Venus, 514 nm/520–540 nm; mCherry, 543 nm/600–620 nm; mRFP, 594 nm/600–620 nm; and propidium iodide (PI), 594 nm/591–636 nm.

All imaging processing was performed with Leica LAS AF Lite and Fiji ImageJ (http://fiji.sc/Fiji) and figures were assembled with Adobe Illustrator 2021. Whenever possible, z-stacked images were obtained. Quantifications and statistical analyses were performed using Fiji and GraphPad Prism, respectively.

**Stomatal phenotype quantification and statistical analysis.** To quantify stomatal index (percentage of stomatal lineage cells), 5-day-old seedlings were stained with propidium iodide (PI, Invitrogen), which enables the visualization of cell outlines, and the adaxial cotyledon epidermal cells were imaged. As described in

ref. [8], cells were counted and classified as belonging to one of three groups: guard cells, pavement cells (cells larger than a mature guard cell and showing at least one obvious lobe), and small dividing cells (round or square cells smaller than a mature guard cell). Stomatal lineage index was calculated as the ratio of the sum of stomatal guard cells and small dividing cells relative to the total number of epidermal cells. To quantify stomatal divisional asymmetry, 4-day-old seedlings of wild type or mutants were stained with PI and imaged on the Confocal to obtain daughter cell sizes in the stomatal lineage. The identify of stomatal lineage cells was aided by the expression of TMM-GFP in wild type and praf4t;5c;8c;9c. Fiji (ImageJ) was used to measure the ratio of surface area of the small daughter cell versus that of the large daughter cell after a cell division.

All statistical analyses were conducted with GraphPad Prism Software. To compare two normally distributed groups, unpaired two-tailed t-tests were used to determine if the difference is significant. For all the figures, *$P < 0.05$, **$P < 0.005$, ***$P < 0.001$, ****$P < 0.0001$ were used.

**Quantification of protein localization in *N. benthamiana* and *A. thaliana*.** To quantify protein polarization in stomatal lineage cells, 10–15 confocal images were taken from the adaxial side of cotyledon epidermis in 3-day-old Arabidopsis seedlings. Protein polarization was measured by the ratios of high fluorescence intensity values over low fluorescence intensity values obtained from equal lengths along the cell periphery and within the same cell. To quantify the number of endosomal compartments (proteins or FM4-64 staining) per cell, 10–15 z-projected confocal images were captured, and the endosomes were counted by Fiji. To quantify the size of vesicles, z-projected confocal images were first processed by "FFT-Bandpass Filter" and "Image-Adjust-Threshold" in Fiji. The sizes of individual vesicles were measured by the function of "Analyze Particles".

For colocalization analysis between PRAF8 and organelle markers in N. benthamiana leaf epidermal cells, 3–5 days after infiltration, z-stacks of sequential scanning images (thickness ~50 μm, with each optical section distanced by 0.5 μm) were captured from the abaxial side of the infiltrated leaves by the Leica SP5 II confocal microscopy. The Fiji Coloc2 plugin was used to obtain the Pearson correlation coefficient (PCC) values using automated thresholding, combined with PSF = 3.0 and Costes randomizations ≥8. Regions-of-interest (ROIs) were circled ($r = 17.05$ μm for 40 smaller ROIs or $r = 34.10$ μm for 10 larger ROIs) from each z-projected image. Only when Costes P-value ≥ 0.95, PCC values above the threshold were recorded and analyzed. To measure the volume size of the BiFC particles, z-projected images (>16 images for each sample) were first deconvoluted by Leica Application Suite Advanced Fluorescence (LAS AF) for 3D analysis (blind deconvolution, 10 iterations). The volume of each particle was measured by Fiji's 3D Object Counter after automatic stack thresholding was applied. The histograms for particle volumes (above 0.01445 $\mu m^3$) were generated by the GraphPad Prism.

**Reporting summary.** Further information on research design is available in the Nature Research Reporting Summary linked to this article.

## Data availability

The data that support the findings of this study are available in the main text or supplementary information. Source data are provided with this paper. The MS dataset is deposited into the MassIVE website with an accession number MSV000088441. All unique biological materials (e.g., plant lines, DNA constructs) are available from the corresponding author upon request. Source data are provided with this paper.

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

## Acknowledgements

We thank Drs. Hidehiro Fukaki and Tatsuaki Goh (Nara Institute of Science and Technology, Japan) for GNOM-related reagents, Dr. Tongda Xu (Fujian Agriculture and Forestry University, China) for His-sumo-BASL, Dr. Hugo H Zheng (McGill University, Canada) for the ST reagents, and Dr. Jiri Friml (IST, Austria) for the *pin* related mutants. We thank the ABRC stock center for providing T-DNA insertional seeds and the WAVE reagents, the Biological Mass Spectrometry Facility of Robert Wood Johnson Medical School for performing MS analysis. This research was supported by grants from the National Institute of Health GM109080 and GM131827, the National Science Foundation 1851907 and 1952823 to J.D. K.Y. is supported by grants from the National Natural Science Foundation of China 31871377 to K.Y. Research in J.L.'s group is supported by grants from the National Natural Science Foundation of China 31771515 and 31970804 to J.L. Research in M.T.M.'s group is supported by Core Research for Evolutional Science and Technology (CREST) award from the Japan Science and Technology Agency (JST) JPMJCR14M5 to M.T.M.

## Author contributions

L.W., D.L., K.Y., and J.D. designed the research. L.W., D.L., and K.Y. performed most of the experiments. X.G. performed co-IP MS of BASL. C.B. contributed to the yeast two-hybrid assays. T.N. assisted with mutant analysis. L.W., D.L., K.Y., J.L., M.T.M., D.C.B., and J.D. wrote the manuscript.

## Competing interests

The authors declare no competing interests.
