## [Peer Review File · Nature Communications]

Connected function of PRAF/RLD and GNOM in membrane trafficking controls intrinsic cell polarity in plantsREVIEWER COMMENTS

Reviewer #1 (Remarks to the Author):

In this manuscript, Wang et al. report four PRAF/RLD proteins as BASL interactors that are required for proper stomatal ACD and BASL polarization. PRAF/RLD proteins were identified from Miyo Morita's group as AtLAZY interactors required for proper PIN polarity during gravitropism response (Furutani et al. 2020 Nature Commun). PRAF/RLDs are localized on cell plasma membrane and post-Golgi endosomes, and show some enriched cell polarity once overexpressed in the stomatal lineage cells. The growth and stomatal development defects phenotype of praf mutants resembles gnom mutants, together with the physical interaction evidence between PRAF and GNOM proteins, suggest a role of PRAF in GNOM-mediated endosomal trafficking.

This is an interesting manuscript reporting important regulators of stomatal-lineage cell polarity, with huge amounts of data provided. The data presented are of high quality and beautifully presented (especially confocal microscopy data). However, the manuscript is disorganized in such a way that one would wonder whether it is reporting: (i) the mechanism for BASL polarization (which underpinning stomatal ACD); (ii) the mechanism of post-Golgi membrane dynamics and recycling by GNOM and PRAF; or (iii) role of GNOM/PRAF on PIN polarity during stomatal development.

The data for detailed characterization of PRAF/RLD subcellular dynamics in the endosomes, using all sorts of WAVE lines, for instance, is interesting. Having said this, since BASL does not localize into endomembrane vesicles (Sup Fig 5c), the whole sets of experiments done (e.g. Fig. 4, Fig. 6) seem irrelevant. Do the authors think that the BFA effects on previous paper (Le et al. 2014 Nature Commun) are due to BASL localization defects, PIN3/auxin defects, or both? Does BASL associate with GNOM? It appears to me that the authors with different expertise (Dong/Bergmann, Le, and Morita) tried to put all information regarding stomatal lineage polarity and BASL, PRAF/RLD, GNOM, PIN, etc, this obscures the take home message of this manuscript.

Specific comments are as the following:

Figure 1:

L66: "We suspected that intramolecular domain folding of the full-length PRAF protein might interfere with their interaction with BASL, or that lipid-binding motifs would make PRAF translocation that makes the assays in the nucleus difficult in yeast."

Why full length of PRAFs do not interact with BASL? Interaction experiments are done with either one PRAF or the other but never shown for all PRAF4/5/8/9. (Supplement Figure1; BifC controls for PRAF 8 missing, PRAF9 interaction in planta missing). Pull down assay with full length PRAF and BASL will be needed since the explanation is not reasonable. Is it justified to use one of PRAFs interchangeably?

Figure2:

L122: "praf4c;5c;8c;9c and praf4t;5c;8c;9c produced extra numbers of stomatal lineage cells."

The author could explain more details about how to identify different cell type (meristemoid vs. SLGC) when counting the cells for divisional asymmetry. It is difficult to understand the definition of cell A and cell B in Fig. 2c, especially for the mutant phenotype with two dividing cell in similar size. Marker such as SPCH (and/or MUTE) as readout?

L138-: The sentence here implies that PRAF is not responsible for the phosphorylation and consequently reorganization to the plasma membrane, however, neither has been shown.

Figure3:

L171: "As GFP-BASL also requires PRAFs to polarize (Fig. 2d, e and Supplementary Fig. 2h, i), the interdependence of PRAF and BASL for polarization suggested a positive feedback loop between the two proteins for the establishment of cell polarity in stomatal lineage cells."

To support the point, the authors need to show PRAF4 behaviors in basl mutant. The whole positive feedback loop is the author's assumption, there are no evidence to support this idea. The author could talk about this later in discussion, but not here in results section.

L182: "Importantly, myr-PRAF8 failed to fully complement the mutant, suggesting the abundant endosomal association of PRAF8 is also essential for its biological function. Thus, we propose that PRAF8 is localized to the plasma membrane and endosomal compartments and PRAF8 functions at both subcellular locations."

The evidence can only support that the mutated version of PRAF8 failed to locate the puncta, as well as to fully rescue the defects. But no further evidence supported there is connection between these two phenotypes.

Figure4:

L206: "Thus, our results revealed that the intracellular PRAF proteins may associate with the endomembrane structures that are sensitive to BFA, such as the Golgi apparatus, TGN/EE, and other post-Golgi/endosomal compartments".

Again, experiments were only done with PRAF8, others are missing. BFA experiment rationale is missing. This part provides opposite evidence to the previous conclusion. Co-localization analyses showed that PRAF does not colocalizes with these compartments. Please provide explanations that reconcile this discrepancy.

Figure 5:

L253: "We found that the stomatal phenotypes of a quintuple *praf4t;5c;8c;9c; gnomT* mutant phenocopied that of the quadruple *praf4t;5c;8c;9c* or *gnom* mutants (Fig. 5c-e), indicating that the four PRAF genes and GNOM might be functionally connected."

The phenocopy results between these higher order mutants of *prafs* and *gnom* cannot fully establish their functional connection. Beyond the physical protein-protein interaction evidence, it is suggested to provide more genetic evidence showing the functional relationship between PRAF and GNOM.

Same question, "similar phenotypes" is not sufficient to support the genetic connection. In this particular case, to support the genetic connection between PRAF and GNOM, testing whether these can rescue each other upon overexpression.

L278: "Moreover, GFP-BASL does not accumulate to the BFA-triggered, FM4-64-positive endomembrane aggregations (Fig. 5g and Supplementary Fig. 5d). Thus, the results led us to propose that, although the polarization process requires PRAF- and GNOM-mediated membrane trafficking, BASL protein per se does not travel with endocytosed vesicles."

What is the author's purpose for this part? This piece of evidence is conflict with the model provided in Figure 7.

Figure 6:

Again, the author failed to explain why suddenly switch to PRAF9 in Fig.6d (nothing is mentioned before).

The data (localisation) is not consistent, and the author do not explain why the post GOLGI localisation matters in Fig.6e-f.

The authors emphasize that PRAF and GNOM localisation in the late Golgi but failed to explain the significance.

Figure 7:

The author incorporates their data wrongfully in the model. The author states that BASL is not part of endosomal recycling machinery (shown with BFA treatment). However, in their model they state that both PRAF and GNOM is also necessary for recycling.

L342: “suggesting that protein homeostasis at the plasma membrane might be disturbed by the PRAF or GNOM mutations.”

It is a bit unclear why protein homeostasis at the plasma membrane suddenly comes in play. As far as I can see, no experiments are conducted to properly address the protein homeostasis.

Minor Comments

l166-168 figure citation does not match the figures.

l155 figure citation does not match with figure.

l187: the wording of FM4-64 binding to the membrane is a vague expression and should be changed.

l238: spelling.

Fig5: WT c-e control is missing.

Fig 6F: negative control missing, magnification is missing on last panel. Localization of PRAF in gnom is missing and vice versa.

SupFig.7 is not mentioned in the manuscript.

Reviewer #2 (Remarks to the Author):

Polarized BASL plays an essential role in the cell polarity that controls stomatal asymmetric cell division, but how the polarity of BASL is established remains largely unknown. This manuscript described an action of a class of PRAF proteins in polarization of BASL. Using yeast 2 hybrid, Co-IP/MS and BiFC, the authors identified an interaction between PRAFs and BASL at the certain side of the plasma membrane. With a combined approach of genetics, biochemistry and microscopy, the authors indicated that PRAFs act together with GNOM in membrane trafficking and such an action of PRAFs and GNOM is crucial for the polarization of BASL. Clearly a step forward in cell biology of stomatal development. The weak link in the manuscript is the localization of the action of PRAFs and thus the proposed model of the action PRAFs.

Major points:

(1) There is no conclusive data that RAB-C1 is on post-Golgi or endosomes, while transient expression of RAB-E1d (Zheng et al., 2005, Plant Cell) and transgenic Arabidopsis expressing RAB-E1d (Speth et al., 2009, Plant Physiol.) indicated that RAB-E1d is largely on Golgi and also partially on the plasma membrane. There is no hard experimental data to support that RAB-E1d is on post Golgi or endosomes claimed (Geldner et al., 2009, Plant J). Note that post-Golgi is in fact a very general term used to refer all endomembrane compartments after Golgi. Known post-Golgi compartments in plant cells at the moment include TGN/EE, PVC/LE and RE.

Based on Fig 3g and 6e, it appears that there are two populations of PRAFs: large and relatively small punctates (Fig 3g and 6e). When transient expressed in tobacco (Fig 4b), it seems PRAFs also have two populations. The authors revealed that in transient expression, there is sometimes very good co-localization of PRAFs with Golgi (large punctates, Fig 4b), sometimes no co-localization at all (small punctates, Figure 4b). To me, this is an indication that those relatively large ones are Golgi (good co-localization with RAB-E1d indeed supports this notion). For small punctates of PRAFs, because PRAFs are also co-localized with internalized FM4-64 and aggregated into BFA bodies (Fig 4a), but not sensitive to WM (sup Fig 3h), it seems that they may represent TGN/EE.

Thus, I strongly suggest the authors re-do their co-localization experiments with ST and RAB-E1d (Golgi); RAB-A1e, Syp61 and VHA-a1 (TGN/EE) using RAB-D1 (mainly on the ER, small portions are on EE, Pinheiro et al. 2009, JCS) RAB-F2a and F2b (PVC/LE). Make sure that the time of transient expression is controlled and same aged leaves are used. It may be that early in transient expression, PRAFs may be largely on Golgi (this could be a reason for different localization with ST reported), in the late stage of expression, PRAF may be largely on TGN/EE. But it's also possible that the difference in co-localization with ST is resulted from different leaf conditions.

(2) It is very interesting that PRAFs interact with GNOM and can change the localization of GNOM from Golgi to some small punctates (Fig 6e). Although GNOM is exclusively localized on Golgi, BFA can move the protein to TGN/EE (Naramoto et al. 2014). Does PRAFs also move GNOM to TGN/EE? Based on Fig 6f

and supplemental figure 6e, the interaction of PRAF8-GNOM occurs at large and small punctates. The large ones can be marked by RAB-E1d, thus, the interaction may be on Golgi. Surprisingly, the signal of BiFC was totally separated from ST-mRFP and mCherry-MEMB12 (Supplemental Figure 6e). However, I noted that the size of BiFC signals in ST-mRFP and MEMB12 was much larger than those in other images. Thus, those BiFC signals with ST-mRFP and MEMB12 may be artificial. I strongly suggest the authors to re-investigate the possibility that those small BiFC signals may represent TGN/EE, and re-do the co-localization with ST-mRFP and MEMB12.

(3) Based on Fig 7a (PIN3 and FM4-64), similar to *gnom*, the *praf* mutant is defective in endosomal morphology (aggregated or patched). It seems not very clear why the authors did not focus on confirming such a morphology of TGN/EE and/or PVC/LE with other markers (perhaps use ER and Golgi markers as controls). This experiment would allow the authors to conclude what roles PRAFs may play in endomembrane trafficking. Note that localization of RAB-C1 is not identified yet, RAB-E1d is likely on Golgi (Fig 7e seems indicate so).

It is very interesting from what I can see that, there are filaments (MTs?) instead of Golgi marked by RAB-E1d in *praf* and *gnom* mutants (less prominent in *gnom*, Fig. 7e), but no much difference in other Rab proteins (A1e, D1, D2a, F2b and C1) examined. If PRAFs can serve as a GEF for RAB-E1d, would it be possible that the recruitment of RAB-E1d to Golgi will be affected in *praf*? It will be interesting to examine what are those filaments to enhance their discussion that PRAF may act as a GEF for certain RABs.

(4) The proposed functional relationship of PRAF and GNOM is very interesting, but lacks of hard evidence yet, thus the authors need to be more careful in the discussion.

Minor points

(1) Line 44, Citation (13) should be added at Guo et al., 2021.

(2) Line 196, Indicate the rationale of using CHX with BFA.

(3) When YFP-PRAF8 was transiently expressed in *Nicotiana Benthamiana* leaves, the distribution/morphology varies in different cells. For example, co-expression with RAB-A1e and RAB-D2a, YFP-PRAF8 appears to have larger punctates than those co-expressed with the other markers (Supplemental Fig 4). Is this because different proteins were co-expressed or different leaf conditions/stages were used in the experiments so that the PRAF8 morphology was different?

(4) The protein size should be labeled in the gel (Fig.6d, Supplemental Fig.1c, and Supplemental Fig.6b).

Reviewer #1:

In this manuscript, Wang et al. report four PRAF/RLD proteins as BASL interactors that are required for proper stomatal ACD and BASL polarization. PRAF/RLD proteins were identified from Miyo Morita's group as AtLAZY interactors required for proper PIN polarity during gravitropism response (Furutani et al. 2020 Nature Commun). PRAF/RLDs are localized on cell plasma membrane and post-Golgi endosomes, and show some enriched cell polarity once overexpressed in the stomatal lineage cells. The growth and stomatal development defects phenotype of praf mutants resembles *gnom* mutants, together with the physical interaction evidence between PRAF and GNOM proteins, suggest a role of PRAF in GNOM-mediated endosomal trafficking.

This is an interesting manuscript reporting important regulators of stomatal-lineage cell polarity, with huge amounts of data provided. The data presented are of high quality and beautifully presented (especially confocal microscopy data). However, the manuscript is disorganized in such a way that one would wonder whether it is reporting: (i) the mechanism for BASL polarization (which underpinning stomatal ACD); (ii) the mechanism of post-Golgi membrane dynamics and recycling by GNOM and PRAF; or (iii) role of GNOM/PRAF on PIN polarity during stomatal development.

We thank the reviewer for spending significant time and effort to evaluate our work. Here, as stated in the Abstract, we mainly report 1) the connected function of BASL and PRAF in the establishment of cell polarity in the stomatal lineage, and 2) the connected function of PRAF and GNOM underlies the polarization of BASL. The reviewer was likely influenced by the established role of GNOM in the regulation of PIN polarity and auxin signaling, which could participate in stomatal development (Le et al., 2014 Nat Comm), but is not the focus of this work. In this revised version, we clarified this point of view in the Discussion, under the section of "Possible functions of PRAF at the plasma membrane".

The data for detailed characterization of PRAF/RLD subcellular dynamics in the endosomes, using all sorts of WAVE lines, for instance, is interesting. Having said this, since BASL does not localize into endomembrane vesicles (Sup Fig 5c), the whole sets of experiments done (e.g. Fig. 4, Fig. 6) seem irrelevant.

We identified the endosomal PRAF/RLD proteins as BASL physical partners and determined that the activity of PRAFs in membrane trafficking is required for the establishment of BASL polarization. Therefore, we believe it is essential to characterize where PRAFs are localized and which trafficking pathway might be important for the establishment of BASL polarity (Fig. 4). Furthermore, we detected physical interaction between PRAF and GNOM, whilst GNOM was reported to predominantly localize to the Golgi apparatus (Naramoto et al., 2014 Plant Cell). Therefore, it was important for us to characterize where the PRAF molecules can possibly interact with GNOM (Fig. 6).

Do the authors think that the BFA effects on previous paper (Le et al. 2014 Nature Commun) are due to BASL localization defects, PIN3/auxin defects, or both? Does BASL associate with GNOM? It appears to me that the authors with different expertise (Dong/Bergmann, Le, and Morita) tried to put all information regarding stomatal lineage polarity and BASL, PRAF/RLD, GNOM, PIN, etc, this obscures the take home message of this manuscript.

Our experiments showed that the BFA treatment disturbed BASL polarization (Fig. 5g) and *gnom* mutants show disturbed PIN3 distribution (Fig. 7a), therefore the BFA effects shall

influence both aspects. However, the stomatal phenotypes of *pin3;4;7* and *pin1;3;4;7* mutants (see below) were found much weaker than those of *basl*, *praf-quad* or *gnom* mutants, suggesting that the stomatal defects in *praf-quad* or *gnom* were likely derived more from defective BASL. See images below (Supplementary Fig. 11d).

Does BASL associate with GNOM? –By using co-IP of GFP-BASL coupled with MS, yeast two-hybrid and BiFC (new data below but not included in the Figures), we did not detect observable interactions between GNOM and BASL.

Specific comments are as the following:

Figure 1:

L66: “We suspected that intramolecular domain folding of the full-length PRAF protein might interfere with their interaction with BASL, or that lipid-binding motifs would make PRAF translocation that makes the assays in the nucleus difficult in yeast.”

Why full length of PRAFs do not interact with BASL? Interaction experiments are done with either one PRAF or the other but never shown for all PRAF4/5/8/9. (Supplement Figure1; BiFC controls for PRAF 8 missing, PRAF9 interaction in planta missing). Pull down assay with full length PRAF and BASL will be needed since the explanation is not reasonable. Is it justified to use one of PRAFs interchangeably?

The full-length PRAF proteins failing to show interaction with BASL in the yeast two-hybrid assays was not tremendously surprising. Furutani et al. 2020 Nature Commun also documented

similar observation that only the subdomains of PRAF/RLD showed interaction with LAZY in yeast cells.

In our *in vitro* pull-down experiments, the expression of full-length PRAF proteins appeared to be toxic to *E. coli* cells, therefore we were not able to produce sufficient proteins for the interaction assays. Probably due to the same reason, Furutani et al. 2020 Nature Commun also used the BRX domain of PRAF/RLD only but not the full-length for the pull-down assays. To address the reviewer's concerns, we now have included all four full-length PRAFs and PRAF_C domains in the BiFC and pull-down assays, respectively, in which positive interactions were detected with BASL (new data, see below and Supplementary Fig. 1c, e).

Figure2:

L122: "praf4c;5c;8c;9c and praf4t;5c;8c;9c produced extra numbers of stomatal lineage cells."

The author could explain more details about how to identify different cell type (meristemoid vs. SLGC) when counting the cells for divisional asymmetry. It is difficult to understand the definition of cell A and cell B in Fig. 2c, especially for the mutant phenotype with two dividing cell in similar size. Marker such as SPCH (and/or MUTE) as readout?

Quantification of divisional asymmetry was mainly based on the sizes of the two daughter cells.

The smaller size of a pair of cells was always used as "A". The mutants defective in stomatal

ACD indeed create daughter cells with similar sizes giving rise to the ratio of A/B close to 1 (as shown in Fig. 2c).

As advised by the reviewer, we have also added the expression of MUTE as readout to indicate the identity of late meristemoids in *praf4t;5c;8c;9c* mutants (new Fig. 2c and Supplementary Fig. 2e). Indeed, the *praf* quadruple mutants, as *basl*, produce two daughter cells that may both express MUTE (see below). Textual changes (tracked) are also shown below.

231 development. Furthermore, the typical divisional asymmetry of the stomatal lineage cells
232 (calculated as ratios of the smaller size A relative to the large size B, Fig. 2c) was disturbed by
233 the *praf* quadruple mutations, to some extent mirroring what was observed in *basl* mutants (Fig.
234 2a, c)⁸. The disruptions of both physical asymmetry and cell-fate asymmetry were found in *basl*
235 mutants. By using the expression of the late meristemoid marker MUTE as readout, which is
236 usually only found in the small daughter in the wild type, was indeed identified in both daughter
237 cells in *praf4t;5c;8c;9c* mutants, as in *basl-2* (Supplementary Fig. 2e). When the four *praf*

L138:- The sentence here implies that PRAF is not responsible for the phosphorylation and consequently reorganization to the plasma membrane, however, neither has been shown.

Yes, indeed we do not know whether the regulation of PRAF on BASL involves protein phosphorylation. To avoid any confusion, we deleted this sentence.

Figure3:

L171: “As GFP-BASL also requires PRAFs to polarize (Fig. 2d, e and Supplementary Fig. 2h, i), the interdependence of PRAF and BASL for polarization suggested a positive feedback loop between the two proteins for the establishment of cell polarity in stomatal lineage cells.”

To support the point, the authors need to show PRAF4 behaviors in *basl* mutant. The whole positive feedback loop is the author’s assumption, there are no evidence to support this idea. The author could talk about this later in discussion, but not here in results section.

We appreciate the reviewer sharing the insights and giving suggestions. We indeed showed the polarization of PRAF8 requires the presence of *BASL* (Fig. 3d-f) but we do agree that elaboration of it is more appropriate for the Discussion. Thus, as advised, the positive feedback model is now moved to the Discussion section.

L182: “Importantly, myr-PRAF8 failed to fully complement the mutant, suggesting the abundant

endosomal association of PRAF8 is also essential for its biological function. Thus, we propose that PRAF8 is localized to the plasma membrane and endosomal compartments and PRAF8 functions at both subcellular locations.”

The evidence can only support that the mutated version of PRAF8 failed to locate the puncta, as well as to fully rescue the defects. But no further evidence supported there is connection between these two phenotypes.

We thank the reviewer for sharing the insights and giving suggestions. The sentences have now been deleted.

Figure4:

L206: “Thus, our results revealed that the intracellular PRAF proteins may associate with the endomembrane structures that are sensitive to BFA, such as the Golgi apparatus, TGN/EE, and other post-Golgi/endosomal compartments”.

Again, experiments were only done with PRAF8, others are missing. BFA experiment rationale is missing. This part provides opposite evidence to the previous conclusion. Co-localization analyses showed that PRAF does not colocalizes with these compartments. Please provide explanations that reconcile this discrepancy.

We have now added new data for PRAF4-YFP, PRAF5-YFP, and PRAF9-YFP treated with BFA (Supplementary Fig. 5a). All the PRAF proteins in the test showed similar responses to the BFA treatment and the wash-out (see below).

As advised, we revised the rationale in the text for the BFA experiment (see below and in Lines 187-193).

“To test whether the PRAF-associated endosomes participate in the endocytic recycling pathway, we further treated the seedlings with Brefeldin A (BFA), an Arf GEF inhibitor that disturbs endomembrane trafficking, leading to the formation of the so-called “BFA-body” compartments that contain aggregated Golgi and trans-Golgi network (TGN)/early endosome (EE) membranes²². Recent study by Qi et al. demonstrated that 30 to 90 μM BFA effectively induces the formation of similarly structured BFA bodies in *Arabidopsis* stomatal lineage cells³⁷.”

In the process of revision, to address some of the Reviewer 2's questions, we further investigated PRAF8's localization to the Golgi and TNG/EE compartments (see details below). The results indeed suggested that PRAF8 associates with both compartments (partial co-localization detected with the Golgi-marker ST and the TGN/EE marker VAMP721) (new data in Fig. 4c). In addition, PRAF8 showed high co-localization with RabE1d that associates with Golgi and endosomes and is sensitive to BFA (Geldner et al., 2009). Furthermore, PRAFs interact with GNOM that is also very sensitive to BFA (Supplementary Fig. 5a). Collectively, these data can now well explain why the PRAF proteins accumulate to the BFA bodies.

Figure 5:

L253: “We found that the stomatal phenotypes of a quintuple *praf4t;5c;8c;9c*; *gnomT* mutant phenocopied that of the quadruple *praf4t;5c;8c;9c* or *gnom* mutants (Fig. 5c-e), indicating that the four PRAF genes and GNOM might be functionally connected.”

The phenocopy results between these higher order mutants of *prafs* and *gnom* cannot fully establish their functional connection. Beyond the physical protein-protein interaction evidence, it is suggested to provide more genetic evidence showing the functional relationship between PRAF and GNOM.

Same question, “similar phenotypes” is not sufficient to support the genetic connection. In this particular case, to support the genetic connection between PRAF and GNOM, testing whether these can rescue each other upon overexpression.

We agree with the reviewer that testing whether PRAF and GNOM upon overexpression can rescue each other would consolidate their genetic connection. Unfortunately, given the restrict revision time frame in particular to deal with these seedling lethal mutants, we were unable to create the desired overexpression lines in the related mutant backgrounds (all our overexpression lines were created with the stomatal lineage specific *TMM* promoter).

To address the reviewer's concerns, we rephrased the statement (see below).

456	2;praf4c;5c;8c;9c (Fig. 2a). Taken together, our phenotypic analyses revealed highly resembling
457	phenotypes of the praf and gnom mutants in overall plant growth and stomatal development. ¶

In addition, we provided new data demonstrating that the *in vivo* localization of PRAF8-YFP and GNOM-GFP were altered in the absence of each other (new data in Supplementary Fig. 9), suggesting that the two proteins are indeed mutually needed for their localization and biological function.

L278: “Moreover, GFP-BASL does not accumulate to the BFA-triggered, FM4-64-positive endomembrane aggregations (Fig. 5g and Supplementary Fig. 5d). Thus, the results led us to propose that, although the polarization process requires PRAF- and GNOM-mediated membrane trafficking, BASL protein per se does not travel with endocytosed vesicles.”
 What is the author’s purpose for this part? This piece of evidence is conflict with the model provided in Figure 7.

We apologize for the confusion. These few sentences have been deleted.

Figure 6:

Again, the author failed to explain why suddenly switch to PRAF9 in Fig.6d (nothing is mentioned before).

The data (localisation) is not consistent, and the author do not explain why the post GOLGI localisation matters in Fig.6e-f.

The authors emphasize that PRAF and GNOM localisation in the late Golgi but failed to explain the significance.

We intended to use PRAF4/8 for the Arabidopsis co-IP experiments with GNOM. However, the protein expression levels of PRAF4/5/8 were relatively low compared with PRAF9 (based on YFP intensity) (new data Supplementary Fig. 8d). We also added the explanation in the text (below).

588 native promoter-driven YFP-tagged PRAF proteins in *Arabidopsis* plants. Because of the low
589 expression levels of PRAF4/5/8-YFP (Supplementary Fig. 8d), we relied on the plants
590 expressing PRAF9-YFP. The co-IP results show that when PRAF9-YFP was pulled down by

It was not explicit to us what the reviewer meant -- “the localization data is not consistent”.

The significance of PRAF8-GNOM interaction occurring at the post-Golgi RabC1 and RabE1 endosomes was discussed later in the Discussion. Although neither of these Rab GTPases has been well-characterized, there were evidence from animals and plants suggesting that they might contribute to targeted secretion/exocytosis for the establishment of cell polarity.

826 *benthamiana* leaf epidermal cells (Fig. 4 and Supplementary Fig. 6). Interestingly, the
827 interactions between PRAF8 and GNOM were identified to mainly occur at the RabC1- and
828 RabE1d-labeled endomembrane compartments (Fig. 6 and Supplementary Fig. 6), emphasizing
829 the relevance or potential importance of the secretion/endosomal recycling in establishing cell
830 polarity. RabC1 has not been well characterized in *Arabidopsis* yet, though its homolog Rab18
831 in mammals was found to associate with the vesicles near the apical surface in polarized
832 epithelial cells to promote targeted secretion^{48,70}. The plant RabE GTPases, homologs of
833 Sec4/Rab8 (yeast/mammals)⁴⁸, were found to localize to the Golgi and promote polarized
834 exocytosis and secretion in both *Arabidopsis* and tobacco^{49,50,71-73}. Thus, we propose that the
835 connected function of PRAF-GNOM may regulate Golgi and post-Golgi endosomal activities,
836 particularly the RabC- and RabE-mediated pathways, to promote directional exocytosis and
837 secretion (Fig. 7g). This hypothesis is supported by the previous observation of cell wall defects

Figure 7:

The author incorporates their data wrongfully in the model. The author states that BASL is not part of endosomal recycling machinery (shown with BFA treatment). However, in their model they state that both PRAF and GNOM is also necessary for recycling.

We appreciate the reviewer sharing the insights. It is likely that our statement in the previous version “BASL is not part of the endosomal recycling machinery” can be confusing or disturbing for the audience.

With regards to the model, our data showed that unless the partner PRAF proteins were overly co-expressed, polarized BASL is localized to the plasma membrane but not to the endosomes, even with the BFA treatment (Fig. 5g). Therefore, BASL was diagramed at the plasma membrane but not at the endosomes. However, indeed we cannot exclude the possibility that a trace amount of BASL molecules bound to PRAFs can exist at the endosomes but were under detection with the current experimental setup. To reconcile this discrepancy, we have deleted the places where we emphasized “BASL is not part of the endosomal recycling machinery” (as described above).

L342: “suggesting that protein homeostasis at the plasma membrane might be disturbed by the PRAF or GNOM mutations.”

It is a bit unclear why protein homeostasis at the plasma membrane suddenly comes in play. As far as I can see, no experiments are conducted to properly address the protein homeostasis.

To avoid unnecessary confusions, as advised, we rephrased “protein homeostasis” to “transmembrane proteins at the plasma membrane undergoing endocytic recycling might be disturbed by the *praf* or *gnom* mutations” (Lines 328-329).

Minor Comments

I166-168 figure citation does not match the figures.

Corrected.

I155 figure citation does not match with figure.

Corrected.

I187: the wording of FM4-64 binding to the membrane is a vague expression and should be changed.

Revised to “we used the styryl dye FM4-64 that intercalates into the plasma membrane, is then taken into the cells by endocytosis” (lines 181-182).

I238: spelling.

Corrected.

Fig5: WT c-e control is missing.

Corrected.

Fig 6F: negative control missing, magnification is missing on last panel. Localization of PRAF in *gnom* is missing and vice versa.

Added.

SupFig.7 is not mentioned in the manuscript.

Corrected.

Reviewer #2 (Remarks to the Author):

Polarized BASL plays an essential role in the cell polarity that controls stomatal asymmetric cell division, but how the polarity of BASL is established remains largely unknown. This manuscript described an action of a class of PRAF proteins in polarization of BASL. Using yeast 2 hybrid, Co-IP/MS and BiFC, the authors identified an interaction between PRAFs and BASL at the certain side of the plasma membrane. With a combined approach of genetics, biochemistry and microscopy, the authors indicated that PRAFs act together with GNOM in membrane trafficking and such an action of PRAFs and GNOM is crucial for the polarization of BASL. Clearly a step forward in cell biology of stomatal development. The weak link in the manuscript is the localization of the action of PRAFs and thus the proposed model of the action PRAFs.

We thank the reviewer for recognizing the value of our work and our new contribution to stomatal cell biology. With regards to the exact localization of PRAF8 and other PRAF members, we believe that a solid definition requires a thorough cell biological investigation that would not be trivial. Previously, the discoveries connecting GNOM-mediated vesicle trafficking to PIN polarization were reported in Steinmann et al., 1999 and Geldner et al., 2003, in which GNOM was hypothesized to localized to the TGN/EE, whereas the localization of GNOM at the Golgi was better characterized in Naramoto 2014, more than 10 years later. On the other hand, this point-of-view (GNOM on Golgi) might be subjected to a change because of the identification of the partially rescuing mutant *gnom^{fwr}* that was mainly localized to the plasma membrane (Okumura et al., 2013), raising the possibility of another functional location of GNOM.

In this story, we put major effort on the characterization of functional connections of PRAF-BASL and PRAF-GNOM, with the intention to provide a brief categorization of where PRAF proteins might be possibly localized. To precisely define where each PRAF member is localized, it will require a tremendous amount of time and effort and should be developed into an independent story. Regardless, we made significant effort in the revision to align our data with the published ones, and conducted the experiments suggested by the reviewer to further test whether PRAF8 localized to the Golgi and TGN/EE. To briefly summarize, we detected partial co-localization with the Golgi markers (ST, RabE1d), the TGN/EE marker (VAMP721) and uncharacterized RabC1 endosomal population. Accordingly, the proposed working model was revised (see below).

Major points:

(1) There is no conclusive data that RAB-C1 is on post-Golgi or endosomes, while transient expression of RAB-E1d (Zheng et al., 2005, Plant Cell) and transgenic Arabidopsis expressing RAB-E1d (Speth et al., 2009, Plant Physiol.) indicated that RAB-E1d is largely on Golgi and also

partially on the plasma membrane. There is no hard experimental data to support that RAB-E1d is on post Golgi or endosomes claimed (Geldner et al., 2009, Plant J). Note that post-Golgi is in fact a very general term used to refer all endomembrane compartments after Golgi. Known post-Golgi compartments in plant cells at the moment include TGN/EE, PVC/LE and RE.

We thank the reviewer for sharing the insights. Indeed, we mainly cited Geldner et al., 2009 to assign the endomembrane compartments in the previous version of the manuscript. In the revision, we have checked throughout the full text to realign how we explain where RabE1d is localized (Golgi) and how PRAF/GNOM may function in membrane trafficking.

Based on Fig 3g and 6e, it appears that there are two populations of PRAFs: large and relatively small punctates (Fig 3g and 6e). When transiently expressed in tobacco (Fig 4b), it seems PRAFs also have two populations. The authors revealed that in transient expression, there is sometimes very good co-localization of PRAFs with Golgi (large punctates, Fig 4b), sometimes no co-localization at all (small punctates, Figure 4b). To me, this is an indication that those relatively large ones are Golgi (good co-localization with RAB-E1d indeed supports this notion). For small punctates of PRAFs, because PRAFs are also co-localized with internalized FM4-64 and aggregated into BFA bodies (Fig 4a), but not sensitive to WM (sup Fig 3h), it seems that they may represent TGN/EE.

Thus, I strongly suggest the authors re-do their co-localization experiments with ST and RAB-E1d (Golgi); RAB-A1e, Syp61 and VHA-a1 (TGN/EE) using RAB-D1 (mainly on the ER, small portions are on EE, Pinheiro et al. 2009, JCS) RAB-F2a and F2b (PVC/LE). Make sure that the time of transient expression is controlled and same aged leaves are used. It may be that early in transient expression, PRAFs may be largely on Golgi (this could be a reason for different localization with ST reported), in the late stage of expression, PRAF may be largely on TGN/EE. But it's also possible that the difference in co-localization with ST is resulted from different leaf conditions.

We highly appreciate the reviewer sharing the insights about the Golgi and TGN/EE behaviors and how PRAFs might possibly associate with them. As advised, we made great effort to re-examine the co-expression pairs of PRAF8, in particular the Golgi and TGN/EE markers, with similarly staged leaves and at different time point (detailed below).

First of all, we expanded our Golgi- and TGN/EE-marker list. We added GONST1 (G) together with the previously tested ST (G), MEMB12 (G), RabE1d (G). We also added additional TGN/EE markers this time, including VAMP721, SYP43, VTI12, SCAMP1, and ECHADNA, together with the previously tested VHAa1 and SYP61. The results showed that co-localizations were detected between PRAF8 with ST, RabE1d, and VAMP721, but not with all the other markers (data not shown). Interestingly, the colocalization with VAMP721 was similar to ST in that two populations of cells (a with $PCC > 0$; b with $PCC < 0$) were found. The new data about PRAF8 co-localization with VAMP721 have been added in Fig. 4c and 4d (see below). Accordingly, we revised text throughout the manuscript suggesting that PRAFs may partially associate with Golgi, TGN/EE, and endosomes.

In addition, we repeated the co-expression experiments as suggested by the reviewer. For each co-expression pair, the images were carefully taken on 2-, 3-, 4-, and 5-day after inoculation. We indeed noticed that the co-localization with the Golgi markers, RabE1d and ST, occurred slightly earlier (2-3 dpi) than with VAMP721 (TGN/EE, 3-4 dpi) in *N. benthamiana* leaf cells.

(2) It is very interesting that PRAFs interact with GNOM and can change the localization of GNOM from Golgi to some small punctates (Fig 6e). Although GNOM is exclusively localized on Golgi, BFA can move the protein to TGN/EE (Naramoto et al. 2014). Does PRAFs also move GNOM to TGN/EE?

Where does PRAF move GNOM to, Golgi, TGN/EE or other endosomes? This is an intriguing question that we also desire the answer.

Co-expression of PRAFs with GNOM induced a drastic change in GNOM localization (from predominant Golgi to smaller punctate). Based on our BiFC results, the interaction between PRAF8 and GNOM was detected to partially overlap with the RabC1- and RabE1d-decorated membrane structures, but not with the TGN/EE markers, such as SYP61, VHAa1, or VAMP721 (new data see below, Supplementary Fig. 10b). However, whether the activity of PRAF8 would move or induce translocation of some GNOM molecules to the TGN/EE is unknown (BiFC only allows visualization of the interaction but not individual protein). To address the reviewer's question, co-expression of a CFP-tagged TGN/EE marker with mCherry-PRAF8 and GNOM-GFP in one *Arabidopsis* plant will be most desired but cannot be obtained within the restricted time frame for the revision.

Based on Fig 6f and supplemental figure 6e, the interaction of PRAF8-GNOM occurs at large and small punctates. The large ones can be marked by RAB-E1d, thus, the interaction may be on Golgi. Surprisingly, the signal of BiFC was totally separated from ST-mRFP and mCherry-MEMB12 (Supplemental Figure 6e). However, I noted that the size of BiFC signals in ST-mRFP and MEMB12 was much larger than those in other images. Thus, those BiFC signals with ST-

mRFP and MEMB12 may be artificial. I strongly suggest the authors to re-investigate the possibility that those small BiFC signals may represent TGN/EE, and re-do the co-localization with ST-mRFP and MEMB12.

We appreciated the reviewer for sharing the insights. We revisited the data of BiFC PRAF8-GNOM co-expression with ST-mRFP or mCherry-MEMB12. The previous images showing seemingly artificial signals are now replaced with the ones showing typical signals (see new Supplementary Fig. 10b).

In the expanded list of Golgi and TGN/EE markers, together with the previously characterized markers Golgi (RabE1d, ST, MEMB12) and TGN/EE (SYP61, VHAA1), we also tested GONST1 (G), SYP43 (TGN/EE), and VAMP721 (TGN/EE) on 2-, 3-, 4-, and 5-dpi, respectively. Unfortunately, we were not able to detect robust co-localization of BiFC PRAF8-GNOM with all of these markers (data not shown), except for RabE1d.

(3) Based on Fig 7a (PIN3 and FM4-64), similar to *gnom*, the *praf* mutant is defective in endosomal morphology (aggregated or patched). It seems not very clear why the authors did not focus on confirming such a morphology of TGN/EE and/or PVC/LE with other markers (perhaps use ER and Golgi markers as controls). This experiment would allow the authors to conclude what roles PRAFs may play in endomembrane trafficking. Note that localization of RAB-C1 is not identified yet, RAB-E1d is likely on Golgi (Fig 7e seems indicate so).

We had to admit that we were not able to finish collecting the whole panel of endomembrane markers, in particular the G and TGN/EE markers, in both *praf* quadruple and *gnom* mutants. Introducing a molecular marker to *praf* quad is particularly tedious because the lethality of the quadruple and severe sterility of the triple mutants. We as the reviewer also very much desire to learn whether the TGN/EE markers are mislocalized in these mutants.

We share the same enthusiasm and desire from the reviewer towards further understanding how the Arf GEF GNOM function together with the poorly understood PRAF proteins in membrane trafficking. Based on the mutant phenotypes, PRAFs and GNOM play functional roles in cell signaling and plant development. We believe additional significant effort is required to combine biochemical, advanced cell biological and genetic strategies to address these big questions.

It is very interesting from what I can see that, there are filaments (MTs?) instead of Golgi marked by RAB-E1d in *praf* and *gnom* mutants (less prominent in *gnom*, Fig. 7e), but no much difference in other Rab proteins (A1e, D1, D2a, F2b and C1) examined. If PRAFs can serve as a GEF for RAB-E1d, would it be possible that the recruitment of RAB-E1d to Golgi will be affected in *praf*? It will be interesting to examine what are those filaments to enhance their discussion that PRAF may act as a GEF for certain RABs.

This was indeed one of the most interesting changes of endomembrane compartments caused by the *praf* and *gnom* mutations. It is possible that RabE1d localization to Golgi requires the

activities of PRAFs and somehow is related to the cytoskeleton. This will require additional makers (MT/actin and Golgi) to be introduced in the quadruple mutants. We have included this suggestion into our to-do-list. The crosstalk between endomembrane compartments with the cytoskeleton is another big interesting topic and how PRAF and GNOM may function in these networks would be an exciting new chapter in our research.

(4) The proposed functional relationship of PRAF and GNOM is very interesting, but lacks of hard evidence yet, thus the authors need to be more careful in the discussion.

We sincerely appreciate the Reviewer's advice. In the revised version, we have deleted the discussion paragraph that might be too speculative.

Minor points

(1) Line 44, Citation (13) should be added at Guo et al., 2021.

Corrected.

(2) Line 196, Indicate the rationale of using CHX with BFA.

Revised as suggested.

(3) When YFP-PRAF8 was transiently expressed in *Nicotiana Benthamiana* leaves, the distribution/morphology varies in different cells. For example, co-expression with RAB-A1e and RAB-D2a, YFP-PRAF8 appears to have larger punctates than those co-expressed with the other markers (Supplemental Fig 4). Is this because different proteins were co-expressed or different leaf conditions/stages were used in the experiments so that the PRAF8 morphology was different?

We revisited the images we have taken for YFP-PRAF8 coexpression lines but did not find a trend of enlarged size by RabA1e or RabD2a. To avoid misinterpretation, we replaced the BiFC RabA2e with another representative image.

(4) The protein size should be labeled in the gel (Fig.6d, Supplemental Fig.1c, and Supplemental Fig.6b).

Revised as suggested.

REVIEWERS' COMMENTS

Reviewer #1 (Remarks to the Author):

The revised manuscript by Wang et al. addressed almost all concerns raised by this Reviewer, who appreciates additional data regarding the BFA treatment of PRAF4/5/9 and MUTE::nGFP analysis. This Reviewer especially enjoyed the revised insightful Discussion.

There are only few minor editorial comments:

1) Lines 82-84. The Y2H experiments are demonstrating that the isolated BASL FxxFxF motif and the PRAF BRX domain are sufficient to interact with each other. Here 'requirement (necessity)' is not tested (the authors test the requirements of these motifs using PRAF_DeltaBRX and BASL_3F->3K using BiFC).

Thus, the sentence should be revised to something like: "Therefore, our data suggest that the physical contact between BASL and PRAF proteins occurs through FxxFxF..." (not "requires").

2) Line 230: Correct to "Furutani et al".

3) P23. Methods, Plant Materials: Please include the origin of MUTE:nGFP (and all additional subcellular TGN/EE LE markers, if possible).

Reviewer #2 (Remarks to the Author):

The revised manuscript is much improved with new data provided and revision of the working model based on new data. There are only two minor issues:

(1) In mammalian cell, Rab18 has been reported to localize to the ER, cis-Golgi, lipid droplets and sometimes endosomes, the localization of RAB-C1 (homolog of Rab18) is not yet experimentally defined in plant cells. Therefore, the authors should not describe the localization of RAB-C1 as 'uncharacterized endosomes', it is better to use e.g. 'uncharacterized compartments' in the text and figures. As such, the authors should thus carefully interpret their co-localization and BiFC data with RAB-C1. It is not quite

correct to interpret the data as co-localization with and/or interaction on some endosomes, respectively.

(2) In my original minor point #3, I referred to YFP-PRAF8 signals in the co-localization of YFP-PRAF8 with RAB-A1e and RAB-D2a (old Supplemental Fig 4, now new Supplemental Fig 6). In the response letter, the authors stated “ We revisited the images we have taken for YFP-PRAF8 coexpression lines but did not find a trend of enlarged size by RabA1e or RabD2a ... ”. However, the images in the mentioned figure were not replaced. In the response letter, the authors stated “... To avoid misinterpretation, we replaced the BiFC RabA2e with another representative image”. Clearly, this was a mistake.

REVIEWERS' COMMENTS

Reviewer #1 (Remarks to the Author):

The revised manuscript by Wang et al. addressed almost all concerns raised by this Reviewer, who appreciates additional data regarding the BFA treatment of PRAF4/5/9 and MUTE::nGFP analysis. This Reviewer especially enjoyed the revised insightful Discussion.

There are only few minor editorial comments:

1) Lines 82-84. The Y2H experiments are demonstrating that the isolated BASL FxxFxF motif and the PRAF BRX domain are sufficient to interact with each other. Here 'requirement (necessity)' is not tested (the authors test the requirements of these motifs using PRAF_DeltaBRX and BASL_3F->3K using BiFC).

Thus, the sentence should be revised to something like: "Therefore, our data suggest that the physical contact between BASL and PRAF proteins occurs through FxxFxF...." (not "requires").

We thank the reviewer for the suggestion. The sentence is revised as suggested.

2) Line 230: Correct to "Furutani et al".

We thank the reviewer for catching the typo. This is now corrected.

3) P23. Methods, Plant Materials: Please include the origin of MUTE::nGFP (and all additional subcellular TGN/EE LE markers, if possible).

The references for the MUTE makers used in the study were added in the text.

Missing description of the additional markers used in the revision was amended in the Methods (Lines 564-570).

Reviewer #2 (Remarks to the Author):

The revised manuscript is much improved with new data provided and revision of the working model based on new data. There are only two minor issues:

(1) In mammalian cell, Rab18 has been reported to localize to the ER, cis-Golgi, lipid droplets and sometimes endosomes, the localization of RAB-C1 (homolog of Rab18) is not yet experimentally defined in plant cells. Therefore, the authors should not describe the localization of RAB-C1 as 'uncharacterized endosomes', it is better to use e.g. 'uncharacterized compartments' in the text and figures. As such, the authors should thus carefully interpret their co-localization and BiFC data with RAB-C1. It is not quite correct to interpret the data as co-localization with and/or interaction on some endosomes, respectively.

We thank the reviewer for the suggestion. We went over all the descriptions about RabC1 and replaced "endosomes" with "compartments/structures" along the text.

(2) In my original minor point #3, I referred to YFP-PRAF8 signals in the co-localization of YFP-PRAF8 with RAB-A1e and RAB-D2a (old Supplemental Fig 4, now new Supplemental Fig 6). In the response letter, the authors stated " We revisited the images we have taken for YFP-PRAF8 coexpression lines but did not find a trend of enlarged size by RabA1e or RabD2a ... ". However, the images in the mentioned figure were not replaced. In the response letter, the authors stated "... To avoid misinterpretation, we replaced the BiFC RabA2e with another representative image". Clearly, this was a mistake.

We apologize for the negligence during the revision. This has now been corrected.